



# A decade (2008-2017) of water stable-isotope composition of precipitation at Concordia Station, East Antarctica

Giuliano Dreossi[1,2], Mauro Masiol[1], Barbara Stenni[1], Daniele Zannoni[1], Claudio Scarchilli[3], Virginia Ciardini[3], Mathieu Casado[4], Amaëlle Landais[4], Martin Werner[5], Alexandre Cauquoin[6], Giampietro Casasanta[7], Massimo Del Guasta[8], Vittoria Posocco[1], and Carlo Barbante[2]

1. Department of Environmental Sciences, Informatics and Statistics, Ca' Foscari University of Venice, Mestre Venice, Italy

2. Institute of Polar Sciences, National Research Council of Italy (ISP-CNR), Mestre Venice, Italy

3. ENEA, Laboratory for Observations and Measures for the Environment and Climate, Rome, 00123, Italy

4. Laboratoire des Sciences du Climat et de l'Environnement, LSCE/IPSL, CEA-CNRS-UVSQ, Université Paris-Saclay, Gif sur Yvette, France

5. Alfred Wegener Institute (AWI), Helmholtz Centre for Polar and Marine Research, Bremerhaven, Germany

6. Institute of Industrial Science, The University of Tokyo, Kashiwa, Japan

7. Institute of Atmospheric Sciences and Climate, National Research Council of Italy (INO-CNR), Bologna, Italy

8. National Institute of Optics, National Research Council of Italy (INO-CNR), Sesto Fiorentino (FI), Italy

*Correspondence to*: Mauro Masiol (mauro.masiol@unive.it) and Barbara Stenni (barbara.stenni@unive.it)

**Abstract.** A ten-year record of oxygen and hydrogen isotopic composition of precipitation is here presented: from 2008 to 2017, 1483 daily precipitation samples were collected all-year round on a raised platform at Concordia Station, East Antarctica. Weather data were retrieved from the Italian Antarctic Meteo-Climatological Observatory AWS, while ERA5 was used to estimate total precipitation. The δ-temperature relationships were moderately high for daily data ($r^2$=0.63 and 0.64 for $\delta^{18}O$ and $\delta^2H$, respectively) and stronger using monthly data ($r^2$=0.82 for both $\delta^{18}O$ and $\delta^2H$), with a slope of about 0.5‰/°C for $\delta^{18}O/T_{AWS}$ (3.5 ‰/°C for $\delta^{18}O/T_{AWS}$), which remains consistent also using annual averages. The isotopic composition of precipitation is the input signal of the snow/ice system and this dataset will be useful to improve the interpretation of paleoclimate records and promote a better understanding of the post-depositional processes affecting the isotopic signal in ice cores. Isotope-enabled GCM ECHAM6-wiso output for the isotopic composition of precipitation was also compared to experimental





data, showing moderately good relationships for $\delta^{18}O$ and $\delta^2H$, but not for d-excess, nonetheless marking a
substantial improvement from the previous release of the model.

## 1 Introduction

Throughout the hydrological cycle, air masses undergo evaporation, condensation, and successive precipitation
events, during which temperature-dependent exchanges of heavy and light isotopes happen due to their slightly
different microphysical properties (Hoefs, 2018). These processes drive the variation of the isotopic composition
of water through all stages of the hydrological cycle and among different reservoirs, e.g., the atmosphere, oceans,
superficial waters, groundwaters, and the cryosphere (e.g., Dansgaard, 1964; Rozanski et al., 1993; Jouzel, 2014).
Consequently, ratios among the three stable isotopes of oxygen ($^{16}O$, $^{17}O$, $^{18}O$) and the two of hydrogen ($^1H$, $^2H$)
have been extensively used as proxies for hydrological, ecohydrological, climatological, palaeoclimatological,
environmental, and agricultural studies from local to global scales (Yoshimura, 2015). Oxygen and hydrogen
isotope ratios are commonly reported as deviations relative to an international standard and are expressed in per
mill (‰):

$$\delta = \frac{R_X - R_{std}}{R_{std}} 10^3 \quad (1)$$

where $R_X$ is either the $^{18}O/^{16}O$ or $^2H/^1H$ ratio in the sample and $R_{std}$ is the same ratio in the V-SMOW standard
(Vienna Standard Mean Ocean Water).
The local temperature at the precipitation site is recognized as the main factor driving the isotopic composition of
precipitation, although occurring progressively over successive condensation events between the initial
evaporation and the final deposition areas. Since the 1950s, a robust relationship between the annual values of the
isotopic composition of precipitations and the average annual local air temperature was reported in mid and high
latitudes (Dansgaard 1953; Epstein and Mayeda 1953; Craig 1961; Dansgaard, 1964; Jouzel et al., 1997, 2003).
In polar regions, this relationship was further supported by theoretical distillation models (Jouzel and Merlivat,
1984) and atmospheric general circulation models (GCMs) (e.g., Risi et al., 2010; Werner et al., 2011).
Besides delta values, the second order parameter deuterium excess (d=$\delta^2H$–8·$\delta^{18}O$; Dansgaard, 1964), provides
additional information on the evaporation conditions at precipitation source regions, i.e., the humidity relative to
saturation during evaporation, the sea surface temperature and, to a limited extent, the wind speed (Merlivat and
Jouzel, 1979; Uemura et al., 2008; Pfahl and Sodemann, 2014; Zannoni et al., 2022). A positive (>0) d-excess is
driven by the higher diffusivity of $^2H^1H^{16}O$ related to $^1H^1H^{18}O$; the result is a relative enrichment of $^2H^1H^{16}O$ in



the vapor-phase during the evaporation process if there is not sufficient time for achieving the isotopic equilibrium between the two phases.

The isotopic composition of surface snow was extensively analyzed in Antarctica, mostly along traverses or close to inland stations. Masson-Delmotte et al. (2008) summarized the available data on the isotopic composition of surface snow across the Antarctic continent. Firn temperature is also usually measured together with snow sample collection as an indicator of mean annual surface temperature (Epstein et al. 1963). These data are extremely important for paleoclimatology: assuming the empirical δ-T relationship valid over time at a specific location, the isotope-temperature slope can be used as an "isotopic thermometer", i.e., to quantify past temperature changes based on the stable isotopic composition. Following this approach, water stable-isotope geochemistry has been widely applied to polar paleoclimate research. The past Earth's climate was reconstructed for over half a century using this approach applied to stratigraphic records of water in ice and firn cores (e.g., Langway, 1958; Gonfiantini and Picciotto, 1959; Dansgaard and Johnsen, 1969; Dansgaard et al., 1993; EPICA Community Members, 2004; Jouzel et al., 2007; Jouzel, 2014; Stenni et al., 2017).

In East Antarctica, where snow accumulation rates are sufficiently low, several deep ice cores recovered over the last decades have provided the reconstructions of past climatic cycles, e.g., 343 kyr (thousands of years) at Talos Dome (Crotti et al., 2021), 420 kyr at Vostok (Petit et al., 1999), 720 kyr at Dome Fuji (Kawamura et al., 2017) and 800 kyr at EPICA Dome C (EPICA community members, 2004; Jouzel et al., 2007). Currently, the European project "Beyond EPICA oldest ice" is underway in the location known as Little Dome C (approx. 35 km from Dome C), aiming to obtain quantitative and high-resolution ice-core information on climate and environmental changes up to 1.5 Myr (https://www.beyondepica.eu/en/).

Major limitations undermine the use of water isotopes for the reconstruction of past temperatures and may bias the interpretation of the paleoclimate records. First, the low snow accumulation rates in inland Antarctica, in combination with wind redistribution effects and stratigraphic noise not related to climate, allow only lower temporal resolutions of ice-core reconstructions compared to high accumulation of coastal regions, generally not finer than decadal or even multidecadal (Petit et al., 1982; Ekaykin et al., 2002; 2004; Frezzotti et al., 2007; Münch et al., 2016; Casado et al., 2018). Another major challenge is linked to the ways the isotopic signal is imprinted and preserved in ice and firn, which is not only shaped by the sensitivity to condensation temperature but also includes further signals of various processes with potentially significant effects on the isotopic fractionation (Casado et al., 2020; 2021). The processes recognized to affect the fractionation processes and mixing during and after the deposition of precipitations were reviewed by Casado et al. (2018) and Ma et al. (2020) and include: (i) processes within the local boundary layer leading to non-constant relationships between the isotopic composition



of snow and surface temperature over time and space (Krinner et al., 1997), (ii) variations in air mass transport trajectories through time (Delaygue et al., 2000; Schlosser et al., 2004), (iii) evaporation conditions at the source

of moisture (Vimeux et al., 1999), (iv) sea surface boundary conditions (Cauquoin et al., 2023); (v) seasonal variations and intermittency of precipitation and accumulation (Touzeau et al., 2016; Casado et al., 2020); (vi) redistribution of snow by surface winds (Groot Zwaaftink et al., 2013; Picard et al., 2019), (vii) water vapor exchanges between surface snow and the atmosphere due to sublimation and solid condensation (Ritter et al., 2016; Genthon et al., 2017); (viii) surface snow metamorphism (Picard et al., 2012; Casado et al., 2021) and (ix)

isotopic diffusion within the firn (Laepple et al., 2018).

These processes are expected to account for large uncertainties in low-accumulation areas, such as East Antarctica, where (i) there is a low precipitation rate (Bromwich et al., 2004; Palerme et al., 2017; Scarchilli et al., 2011; Casado et al., 2018;2020), (ii) the atmospheric dynamics and pathways and surface mass balance (SMB) are yet to be fully understood (Frezzotti et al., 2004;2007; Urbini et al., 2008; Scarchilli et al., 2010;2011), and (iii) the

snow surface remains exposed to the atmosphere for a long time, allowing prolonged interactions and exchanges at the snow-atmosphere interface, longer mixing and potential horizontal transports by winds. Although all these processes are potentially able to bias the pristine isotopic signal of precipitation, their effects on driving the isotopic composition of Antarctic precipitation are still unclear and have not yet been fully quantified. Consequently, the different sensitivity of the empirical δ-T relationship in East Antarctic ice is generally poorly

constrained with respect to other regions (Sime et al., 2009; Stenni et al., 2017; Münch et al., 2016). Because of these limitations, a nonconstant relationship between the snow isotopic composition and air temperature through time and space is expected, as already evidenced by Masson-Delmotte et al. (2008). Other studies further suggest that the δ-T in East Antarctica may vary among ice core sites, with the climatic signal expected to account for only 10%–50% of the variance in $\delta^{18}O$ (Münch and Laepple, 2018; Laepple et al., 2018; Casado et al., 2020;

2021). The three years monitoring (Jan 2008-Dec 2010) of daily precipitation collected at the Concordia Station in the East Antarctic plateau showed clear relationships between the isotopic composition and local air temperature at daily ($R^2$=0.63) and monthly ($R^2$=0.82) scales (Stenni et al., 2016). However, the temporal relationship between daily $\delta^{18}O$ and air temperature was approximately 2-folds smaller than the average Antarctic spatial relationship obtained by Masson-Delmotte et al. (2008), i.e., the one used for the interpretation of EPICA Dome C record

(0.49‰/℃ vs. 0.8‰/℃).

Under this view, there is a need for a better understanding of how the isotopic composition of water is imprinted in the fresh snow and firn and how it evolves to obtain robust and unbiased empirical relationships between climate and stable water isotope signatures. Since the preliminary study by Stenni et al. (2016), the precipitation collection





at Concordia Station for analysing the isotopic composition has continued until the present day and it is still
ongoing. Here, results spanning over 10 years (2008-2017) are reported and discussed. This dataset represents an
unprecedentedly long record of precipitation experimentally measured in East Antarctica with the potential to
better constrain the δ-T thermometer, as well as to better understand the effects of post-depositional processes
since the precipitation isotopic composition is the input signal of the atmosphere-snow surface and subsurface
system. Results are also compared with simulated values of the up-to-date isotope-enabled general circulation
models (iGCMs) ECHAM-wiso providing high temporal resolution isotope records in precipitation.

## 2 Materials and methods

### 2.1 Precipitation collection

Concordia Station (75°06'S, 123°21'E; elevation 3233 m a.s.l.) is a French–Italian research facility located at
Dome C on the East Antarctic Plateau (Figure S1, http://www.concordiastation.aq) open all year round since 2005.
The sampling site is located in the clean area, approx. 800 m from Concordia Station, to avoid contamination from
the anthropogenic operations. Precipitation accumulates over an 80x120 cm wooden platform standing 1 m above
the snow surface. The platform is covered by a polystyrene/polytetrafluoroethylene surface and is shielded by an
8-cm rail to prevent snow from being blown off from the surface. Samples were manually collected with daily
frequency by removing all the accumulated material, which was immediately sealed into labelled plastic bags.
Bags were preserved in a frozen state until the analysis. If no snow or a too low amount of snow was found on the
plate, no sample was collected, and the plate was cleaned. The amount of deposited snow varied depending on the
amount of precipitation from 0 to ~10 mm, with isolated cases of 30–50 mm deposition possibly related to blowing
snow events. The sample is therefore representative of a fresh snowfall, but it may also include snow blown onto
or off the platform by winds. Every day, the collection of the samples was recorded in a logbook reporting the
timing and some weather variables. Generally, the sample collection occurred in the morning between 9 am and
noon, local time (UTC+8).

### 2.2 Analytical

Once in the lab, samples into the sealed bags were melted at room temperature, transferred into 25 mL high-
density polyethylene capped bottles, and then stored at -20°C until analysis. The isotopic composition ($\delta^{18}$O and
$\delta^2$H) of the samples was determined by the well-established $CO_2$-$H_2$/water equilibration method adapted from
Epstein and Mayeda (1953) and Horita (1988), followed by isotope ratio mass spectrometry (IRMS) analysis.





IRMS was composed of a Thermo-Fisher Delta Plus Advantage mass spectrometer coupled with an automatic equilibration device (Finnigan MAT HDO 1086).

Since IRMS requires at least 4 mL water volume, samples having less than 5 mL of melted snow were directly
analyzed without any pre-processing by cavity ring-down spectroscopy (CRDS). CRDS analysis was performed with PICARRO model L1102-*i* and model L2130-*i* equipped with a A1102 vaporizer device. Between-sample memory effects may bias CRDS analyses (Penna et al., 2012), therefore samples were injected 8 times and results were filtered using an outlier test, i.e., discarding all the results falling outside of the interval described by the average of the 8 repetitions ±standard deviation.

All data were expressed as relative to the international standard V-SMOW. Data consistency between analytical methods was assured by several laboratory tests carried out to detect possible biases due to the use of IRMS or CRDS. Average differences were in the order of analytical precision of IRMS (better or equal to ±0.05‰ for $\delta^{18}O$ and ±0.7‰ for $\delta^2H$) and the analytical precision for CRDS (±0.10‰ for $\delta^{18}O$ and ±0.5‰ for $\delta^2H$).

**2.3 Weather data**

Weather data measured at Concordia Station were retrieved from the automatic weather station (AWS) Concordia (WMO ID: 89625), managed by the Italian Antarctic Meteo-Climatological Observatory (Grigioni et al., 2022) and operating since 2005. AWS data include air temperature ($T_{AWS}$, °C), pressure ($Press_{AWS}$, hPa), relative humidity ($RH_{AWS}$, %), wind speed ($ws_{AWS}$, m s$^{-1}$) and direction ($wd_{AWS}$, degrees). Missing hourly AWS data (8.5 %) were reconstructed through linear regression using data measured at the nearby AWS "Dome C II" (WMO ID:
89828), an American station installed in 1995 by the Antarctic Meteorological Research Center (AMRC). The coefficient of determination between AWSs was $r^2=0.99$ for air temperature and surface pressure, $r^2=0.70$ for wind speed. Surface-based temperature inversions (SBTIs) frequently occur within the atmospheric boundary layer across continental Antarctica (Connolley, 1996; Pietroni et al., 2014). At Concordia, strong and long-lived SBTIs are generally observed, reaching up to 40°C in winter and mostly extending within the lowest 100 m of
height, while they may disappear only in the early afternoon during summer due to maximum insolation and convective mixing (Genthon et al., 2010; Argentini et al., 2014; Petenko et al., 2019). Since the condensation temperature can be approximated to the temperature at the upper limit of the inversion layer (Masson-Delmotte et al., 2008), data from daily radiosounding profiles were processed to determine the temperature at the bottom of the first layer where temperature decreases with altitude ($T_{INV}$). The inversion strength (I) was calculated as the
difference between $T_{INV}$ and $T_{AWS}$ (Connolley, 1996). Data of solar direct radiation (Direct rad$_{BSRN}$, W m$^{-2}$) measured at Concordia station were retrieved from the Baseline Surface Radiation Network (BSRN), a network



of the Work Climate Research Program (WCRP) (Ohmura et al., 1998; Driemel et al., 2018; Lupi et al., 2021; Bai et al., 2022).

Reanalysis meteorological data were retrieved from the European Centre for Medium-Range Weather Forecasts
(ECMWF) ERA5 (Hersbach et al., 2023); data of 2m temperature ($T_{2m\ ERA5}$), surface pressure ($Press_{ERA5}$), total precipitation ($tp_{ERA5}$), evaporation ($e_{ERA5}$), cloud base height data ($cbh_{ERA5}$) were downloaded from the Copernicus Climate Change Service (C3S) Climate Data Store (CDS). Relative humidity ($RH_{ERA5}$) was computed from vapor pressure and saturation vapor pressure according to the Murphy and Koop (2005) formulae and using hourly ERA5 data.

The AWSs, BSRN and ERA5 provide meteorological variables with different frequencies (minutes to hours); daily averages were calculated either relative to the local time for investigating the daily patterns or relative to the exact extension of the sampling time for the fine match with the isotopic composition of precipitation, i.e., referring to the information in the logbook. Since missing data can affect the analysis, daily averages were computed only for days having at least 75% of the available hourly records; monthly averages/medians for months
having at least 75% of the available days. When used along with the isotopic composition of snow, monthly and annual-averaged weather data were computed only over days with available samples.

The Southern Annular Mode (SAM, aka Antarctic Oscillation, AAO) depicts changes in the position and strength of the westerly wind belt over the Southern Ocean and is defined as the zonal mean atmospheric pressure difference between the mid-latitudes (~40° S) and Antarctica (~65° S) (Thompson and Wallace, 2000; Marshall,
2003). SAM is the predominant atmospheric variability mode in the Southern Hemisphere having important impacts on temperature and precipitation, including in Antarctica (Fogt and Marshall, 2020). Positive SAM phases lead to cool and dry conditions across the continental Antarctic continent and warm and wet conditions over the Antarctic Peninsula. Daily and monthly SAM indexes were retrieved from the Climate Prediction Center, National Centers for Environmental Prediction of NOAA.

In this study, four meteorological seasons are used: Austral summer (December, January, and February [DJF]), autumn (March, April, and May [MAM]), winter (June, July, August [JJA]), and spring (September, October, November [SON]).

**2.4 iGCMs**

Since 1980s, several isotope-enabled general circulation models (iGCMs) have been developed with explicit
diagnostics for the isotopic composition of water, e.g., NASA GISS, ECHAM-wiso, GENESIS, LMDZ-iso4, iCAM5, MIROC5-iso (Joussaume et al., 1984; Jouzel et al., 1987; Hoffmann et al., 1998; Mathieu et al., 2002;



Schmidt et al., 2005, Risi et al., 2010; Nusbaumer et a., 2017). By incorporating physical processes influencing the isotopic composition of water at all water bodies and at all stages of the water cycle, iGCMs return the isotopic composition in precipitation, water vapor, and snow/ice.

Recently, ECHAM5-wiso (Werner et al., 2011) and ECHAM6-wiso (Cauquoin et al., 2019;2021) have been developed, based on ECHAM5 (Roeckner et al., 2003) and ECHAM6 (Stevens et al., 2013) models, respectively. For both ECHAM-wiso model releases, a nudged simulation was performed that covers the time period of the available isotope measurements at Concordia Station. Reanalysis data from ECMWF were used as input for nudging the iGCM: ERA-Interim Reanalysis data (Dee et al., 2011) for ECHAM5-wiso, ERA5 (Hersbach et al.,

2020) for ECHAM6-wiso. ECHAM-wiso data from both nudged simulations were extracted at the nearest grid cell from Concordia Station, providing modeled daily-averaged values for the temperature at 2 m ($T_{2m\ ECHAM(5,6)}$), surface temperature ($T_{surf\ ECHAM(5,6)}$), the amount of precipitation, $\delta^{18}O$, $\delta^{2}H$, and d-excess values.

## 2.5 Data processing

Statistical and geostatistical analyses were performed using R 4.2.2 (R Core Team, 2022) and a number of

packages, including "boot" (Canty and Ripley, 2022), "bootstrap" (Leisch, 2019), "car" (Fox and Weisberg, 2018), "caret" (Kuhn, 2022), "corrplot" (Wei and Simko, 2021), "DAAG" (Maindonald and Braun, 2011), "mgcv" (Wood, 2004;2011;2017; Wood et al., 2016), "ncdf4" (Pierce, 2023), "rcompanion" (Mangiafico, 2022), "zoo" (Zeileis and Grothendieck, 2005).

Many simple and multiple linear regression analyses were performed on the data for assessing the relationships

between variables. Besides the linear models, the 95th confidence intervals (c.i.) in the prediction of the coefficients (slopes and intercept) were assessed by ordinary nonparametric bootstrap resampling (Davison and Hinkley, 1997) over at least R=2000 replicates (R larger than the input observations). The measure of performance and predictive ability of regression models were also estimated by *k*-fold cross-validation (Maindonald and Braun, 2011; James et al., 2013). This technique randomly partitions the datasets into *k* (*k*=5, in this case) equal-sized

subsamples and recursively uses *k*-1 subsets of the observations to refit the regression while using the remaining part as a testing set. The root mean squared error (RMSE) and mean absolute error (MAE) were computed from residuals of original and cross-validated models as a quantitative measure of errors associated with the estimates. Trends and seasonal patterns of variables were quantified by adopting different approaches applied to the monthly-averaged data for months having at least 75% of the available records. The presence of statistically significant

long-term (monotonic) linear trends during 2008-2017 was assessed through the Theil-Sen nonparametric





estimator of slope (Theil, 1950; Sen, 1968) along with the Mann-Kendall test for trend (Mann, 1945; Kendall, 1975).

## 3. Results and Discussion

### 3.1 Weather and boundary layer dynamics

The time series and monthly/daily patterns of meteorological variables recorded from the AWS or modeled by ERA5 are shown in Fig. S2 and S3. The full period (2008-2017) average TAWS was -53°C with daily averages ranging from -82°C to -19°C and hourly values varying from -83.6°C to -14.3°C. A very high agreement between daily TAWS (blue) and T2m ERA5 (pink) was found ($r^2$=0.95). The East Antarctic Plateau is characterized by strong surface temperature inversions (Baas et al., 2019), which exhibited the same seasonal pattern of surface air

temperature and ranged from -75 to -11°C. The daily inversion strength, calculated between daily radiosounding profiles and TAWS varied from -6 to 48°C (Figure SI2), and its temporal pattern is the mirror image of the air temperature, with higher values during the austral winters. During the coldest months (April to September), the inversion strength generally exceeds 20°C, while is less than 10°C in austral summer (Figure S3) because of the erosion due to the diurnal cycle of solar radiation.

Hourly relative humidity (RH) measured by the AWS varied from 6% to 84% (full period average 46%) with a seasonal pattern similar to air temperature. However, Genthon et al. (2013; 2017) reported frequent supersaturation events not detected by commercially available sensors. Thus, the atmospheric moisture on the Antarctic Plateau could probably be underestimated. Under this view, the relative humidity over ice (RHi) calculated from hourly ERA5 data ranged from 42 to 100% (average 64%). Figure S2 also exhibits data of RHi

experimentally obtained by Genthon et al. (2017) for 2015 using hygrometry sensors modified for air sampling without artifacts. Results show hourly RHi in the 51-131% range (average 89%) with approx. 23% of 2015 over RHi 100%.

Wind roses calculated on a seasonal basis (Figure S4) show prevailing winds from the 4th quadrant (SW-S) throughout the year, peaking from the South, i.e., blowing from the highest plateau and inner regions of Antarctica.

Although possible instrumental issues due to frost deposition may had led to an underestimation of wind at lower speeds, the wind blew in the 0 to 20 m s-1 interval (full period average: 3 m s-1), with slightly higher values in November and around noon. These patterns are consistent with the literature for Dome C (e.g., Argentini et al., 2014).





Total precipitation and solar irradiation were not measured by AWS; in this study, values from ERA5 were used
for total precipitation, while BSRN data were used for solar radiation. The annual cumulative amount of total
precipitation during 2008-2017 ranged from approx. 20 to 30 mm y-1 (average 24 mm y-1), in accordance with
previous ERA-Interim data (1979–2012, Genthon et al., 2016). Monthly (Figure S2), precipitations were roughly
constant throughout the year with slightly lower values between October and December. The diel pattern was also
quite flat, with lower values around noon. During the 2008-2017 period, the hourly average solar direct radiation
was in the 0-1175 W m-2 range (average 345 W m-2). The monthly and hourly cycles (Figure S3) clearly reflect
the solar elevation patterns with the highest values recorded during the Austral summer and midday hours.

SAM during 2008-2017 was predominantly positive, as reported in previous studies (Fogt and Marshall, 2020).
Six main negative periods counting at least 3 continuative months with SAM index <0 were identified (May-Sept.
2009; June-Sept. 2011; Oct. 2012-Jan. 2013; Aug.-Oct. 2013; Aug.-Nov. 2014; Oct. 2016-Jan. 2017). Simple
cross-correlations between the SAM index against air temperature from the AWS show only statistically
significant small negative correlations at negative lags only using daily data. Results indicate that a higher daily
SAM index is generally related to decreasing air temperatures after 2/3 days (all periods, DJF, MAM, JJA) and
8/9 days (SON) (Figure S5).

### 3.2 Water stable isotope data

A total of 1483 daily samples were collected at Concordia Station and analyzed for the oxygen and hydrogen
isotopic composition of snow. Thus, samples were collected over ~41% of days in 10 years. On a monthly basis,
no precipitation was collected for 3 months (November 2009 and 2014, December 2015); approximately one third
of the months include at least one week of available samples, while samples were collected on at least 90% of the
days for 9 months (Figures S6 and S7). $T_{AWS}$ during the days with collected samples was slightly (1.6°C) but
significantly (Wilcoxon rank sum test with continuity correction $p<0.05$) lower during the sampling days with
respect to non-sampling days; at a seasonal basis, the $T_{AWS}$ difference was -2.5°C.

Figure 1 shows the time series of the stable isotope composition data along with the air temperature measured by
the AWS. $\delta^{18}O$ varied between -82.63 and -26.97‰ (average -56.7‰); $\delta^2H$ varied between -595.1 and -223.0‰
(average -438‰). The minimum delta values are amongst the most isotopically depleted waters collected on the
Earth so far. Violin plots in Fig. S8 and S9 show that data distributions were quite symmetrical (median -56.8‰
and -440‰ for $\delta^{18}O$ and $\delta^2H$, respectively).



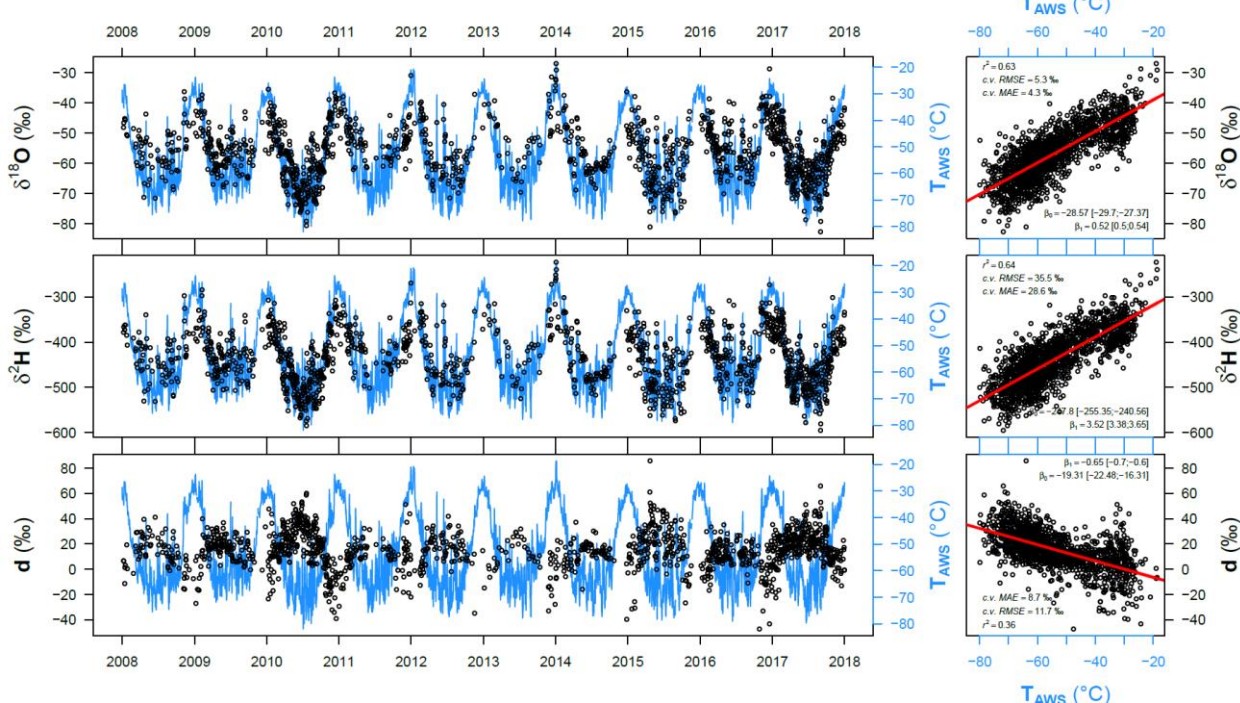

**Figure 1. Left: time series of the daily averaged air temperature (blue lines) and the isotopic composition of snow (black dots) measured at Concordia during 2008-2017. Right: linear regressions between the isotopic composition of snow and air temperature.**

Besides the monthly arithmetic averages, isotope data and air temperature are also computed as precipitation-weighted averages using the total precipitation amount from ERA5 (reported as $\delta^{18}O_{tp}$; $\delta^2H_{tp}$; $d_{tp}$; $T_{tp}$). The monthly patterns (Figure 2) closely followed the air temperature: the lower delta values were generally recorded during the Austral winters, reflecting the "temperature effect", i.e., the positive relationship between the isotopic composition of precipitation and air temperature mainly observed at high and mid-high latitudes (Dansgaard, 1964; Rozanski et al., 1993). Generally, the lower monthly-averaged delta values were measured during June ($\delta^{18}O$=-64.2‰; $\delta^2H$=-486‰) and the higher values in January ($\delta^{18}O$=-46.2‰; $\delta^2H$=-365‰). The average seasonal amplitude of delta values between DJF and JJA spanned over approx. 16‰ and 111‰ for $\delta^{18}O$ and $\delta^2H$, respectively.

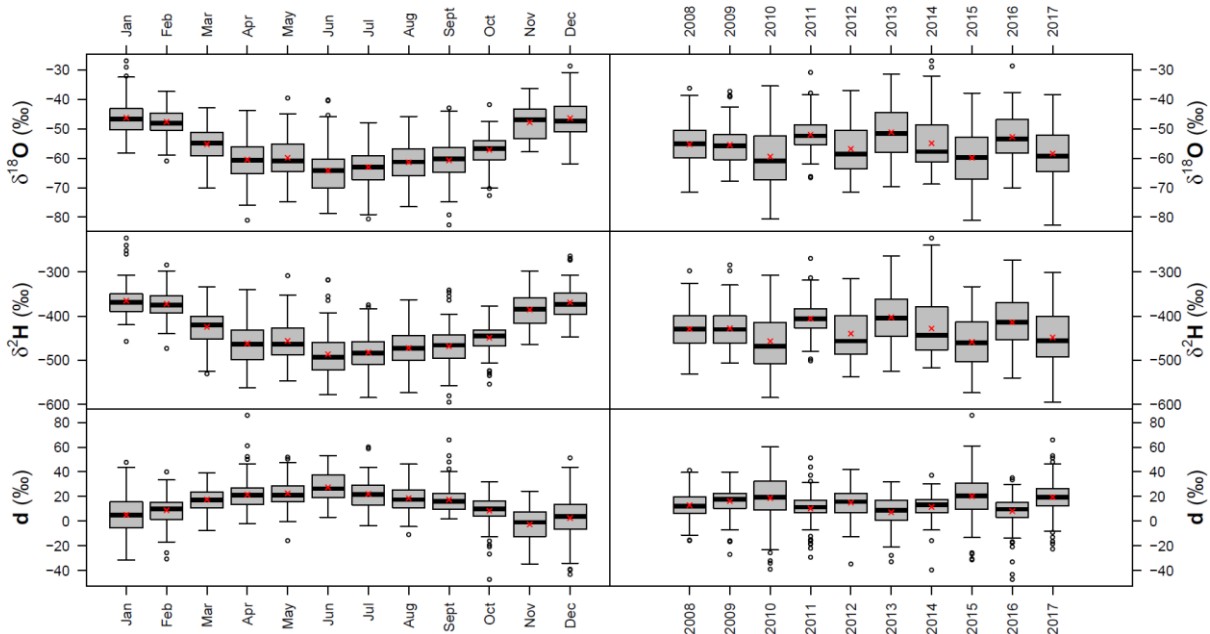

**Figure 2. Seasonal (left) and annual (right) boxplots of the stable isotope composition of snow collected at Concordia station (line = median, box = inter-quartile range, whiskers=±1.5*inter-quartile range, circles = outliers and extremes; red crosses = arithmetic mean).**


The temperature effect upon the isotopic composition of precipitation is evident. Delta values exhibited a strong seasonality (Figure S10) with the less negative values recorded during Austral Summers. Under this view, the simple linear relationship of daily values of $\delta^{18}$O and $\delta^2$H with $T_{AWS}$ was moderately high ($r^2$=0.63 and 0.64, respectively) (Figure 1; Table 1). The regression slopes were 0.52‰/°C [cross-validated 95% c.i.: 0.50-0.54‰/°C]

and 3.52‰/°C [3.38-3.65‰/°C] for $\delta^{18}$O and $\delta^2$H, respectively. These positive relationships become stronger on monthly-averaged data ($r^2$=0.82 for both $\delta^{18}$O and $\delta^2$H) and slopes of 0.51 [c.i.: 0.46-0.55‰/°C] and 3.4‰/°C [c.i.: 3.09-3.73‰/°C] for $\delta^{18}$O and $\delta^2$H, respectively (Figure S11; Table 1). Similar results were also obtained by regressing delta values against $T_{AWS}$ weighted for ERA5 tp: $\delta^{18}O_{tp}$ (0.52‰/°C) and $\delta^2H_{tp}$ (3.56‰/°C) against $T_{AWS\ tp}$ (Figure S12; Table 1). Finally, regressions computed over annually-averaged data also exhibited high

coefficients of determination and similar slopes (Figures S13 and S14). At annual basis, regression slopes were





slightly higher:0.59‰/°C [cross-validated 95% c.i.: 0.39-0.83‰/°C] and 3.9‰/°C [2.8-5.5‰/°C] for $\delta^{18}$O and $\delta^2$H, respectively.

**Table 1. Results of the regressions of delta values against air temperature.**

| Regression variables | | Data used | Intercept | | Slope | | $r^2$ | CV RMSE |
| Dependent | Independent | | $\beta_0$ (±std. error) | BS 95th CI | $\beta_1$ (±std. error) | BS 95th CI | | ‰ |
|---|---|---|---|---|---|---|---|---|
| $\delta^{18}$O | T | Daily | -29 (±1) | [-30;-27] | 0.52 (±0.01) | [0.5;0.54] | 0.63 | 5.34 |
| $\delta^2$H | T | Daily | -248 (±4) | [-255;-241] | 3.52 (±0.07) | [3.38;3.65] | 0.64 | 35.5 |
| d | T | Daily | -19 (±1) | [-22;-16] | -0.65 (±0.02) | [-0.7;-0.6] | 0.36 | 11.7 |
| $\delta^{18}$O | T | Monthly | -29 (±1) | [-31;-26] | 0.51 (±0.02) | [0.46;0.55] | 0.82 | 3.05 |
| $\delta^2$H | T | Monthly | -251 (±8) | [-268;-232] | 3.4 (±0.15) | [3.09;3.73] | 0.82 | 20.5 |
| d | T | Monthly | -21 (±3) | [-29;-14] | -0.65 (±0.05) | [-0.78;-0.52] | 0.57 | 7.5 |
| $\delta^{18}$O$_{tp}$ | T$_{tp}$ | Weighted monthly | -27 (±1) | [-30;-24] | 0.52 (±0.02) | [0.47;0.58] | 0.81 | 3.11 |
| $\delta^2$H$_{tp}$ | T$_{tp}$ | Weighted monthly | -239 (±8) | [-257;-218] | 3.56 (±0.17) | [3.2;3.95] | 0.79 | 22.5 |
| d$_{tp}$ | T$_{tp}$ | Weighted monthly | -21 (±3) | [-29;-15] | -0.63 (±0.05) | [-0.77;-0.52] | 0.55 | 7.1 |
| $\delta^{18}$O | T | Annual | -24 (±6) | [-34;-12] | 0.59 (±0.12) | [0.39;0.83] | 0.75 | 1.6 |
| $\delta^2$H | T | Annual | -223 (±41) | [-280;-140] | 3.9 (±0.77) | [2.78;5.5] | 0.76 | 10.3 |
| d | T | Annual | -30 (±13) | [-56;-4] | -0.82 (±0.23) | [-1.3;-0.33] | 0.61 | 3.2 |
| $\delta^{18}$O$_{tp}$ | T$_{tp}$ | Weighted annual | -22 (±6) | [-40;-9] | 0.62 (±0.14) | [0.24;0.93] | 0.71 | 1.7 |
| $\delta^2$H$_{tp}$ | T$_{tp}$ | Weighted annual | -188 (±50) | [-338;-81] | 4.52 (±1.08) | [1.25;6.88] | 0.69 | 13.9 |
| d$_{tp}$ | T$_{tp}$ | Weighted annual | -13 (±10) | [-30;6] | -0.48 (±0.22) | [-0.84;-0.06] | 0.37 | 2.4 |


### 3.3 Local meteoric water lines

The local meteoric water line (LMWL) reveals the linear relationship between $\delta^{18}$O and $\delta^2$H (Craig, 1961; Dansgaard, 1964). LMWLs were computed by considering all single samples from the entire study period (Figure 3) as well as using aggregated data for the entire dataset or over each season (Figures S16-S19). Regression statistics are also summarized in Table 2; the regression coefficients were always statistically significant (*p*<0.05). The LMWL computed over daily data was ($r^2$=0.98):

$$\delta^2H = 6.65 \ [6.59;6.71] \cdot \delta^{18}O - 60.72 \ [-64.3;-57.1]$$




with a prediction error for $\delta^2$H of 8.4‰ (5-folds cross-validated RMSE); these results are very similar to the values for the 2008-2010 period ($\delta^2$H = 6.5 · $\delta^{18}$O – 68.8) reported by Stenni et al. (2016). The LMWL computed over monthly-averaged data weighted for ERA5 total precipitation was:

$$\delta^2 H_{tp} = 6.83\ [6.64;7.07] \cdot \delta^{18}O_{tp} - 52.29\ [-62.43;-37.81]$$

The intercepts ($\beta_0$) and slopes ($\beta_1$) of all LMWLs computed with different data and periods are summarized in Fig. 4 along with their cross-validated c.i. Generally, the coefficients of the regressions for the weighted and unweighted monthly data return very similar results over the 4 seasons when accounting for the confidence intervals (Table 2). On the contrary, the LMWLs calculated at a seasonal basis (Figure 4, color symbols) generally
show lower slopes (6.02 [c.i.: 5.73;6.32]) and intercepts (-86.76 [c.i.: -101;-72]) in Austral summers, while higher regression coefficients were recorded in autumns (slope 6.72 [6.58;6.82], intercept -54 [-63;-49]), clearly depicting the seasonal effect over the isotopic composition. Monthly LMWLs on weighted data show slightly higher intercepts and slope values than the daily and unweighted monthly data (Table 2). In all cases, summer
LMWLs show lower $r^2$ compared to the other seasons as well as to the entire dataset.
The slope of the LMWL exhibits lower values than the 7.75 obtained by Masson-Delmotte et al. (2008) for the whole Antarctic surface snow database, although in better agreement with the one obtained for the last quartile of the isotopic distribution and corresponding to 7.28, for isotopic $\delta^{18}$O values of surface snow below -42.8‰. This reflects the lower slope of the MWL when dealing with very depleted precipitation at the final stages of the
isotopic distillation line. In agreement with theoretical isotopic models (Jouzel and Merlivat, 1984), the MWL slopes in surface snow of East Antarctica decrease from the coastal areas to the inland plateau (Masson-Delmotte et al., 2008). However, when considering the MWL calculated on the annual average data weighted for ERA5 total precipitation, the slope (7.33) is in very good agreement with the one reported from Masson-Delmotte et al. (2008). Indeed, the surface snow data consider the first few meters of the snowpack, which corresponds, in theory,
to the precipitation of several years already "naturally" weighted for precipitation.



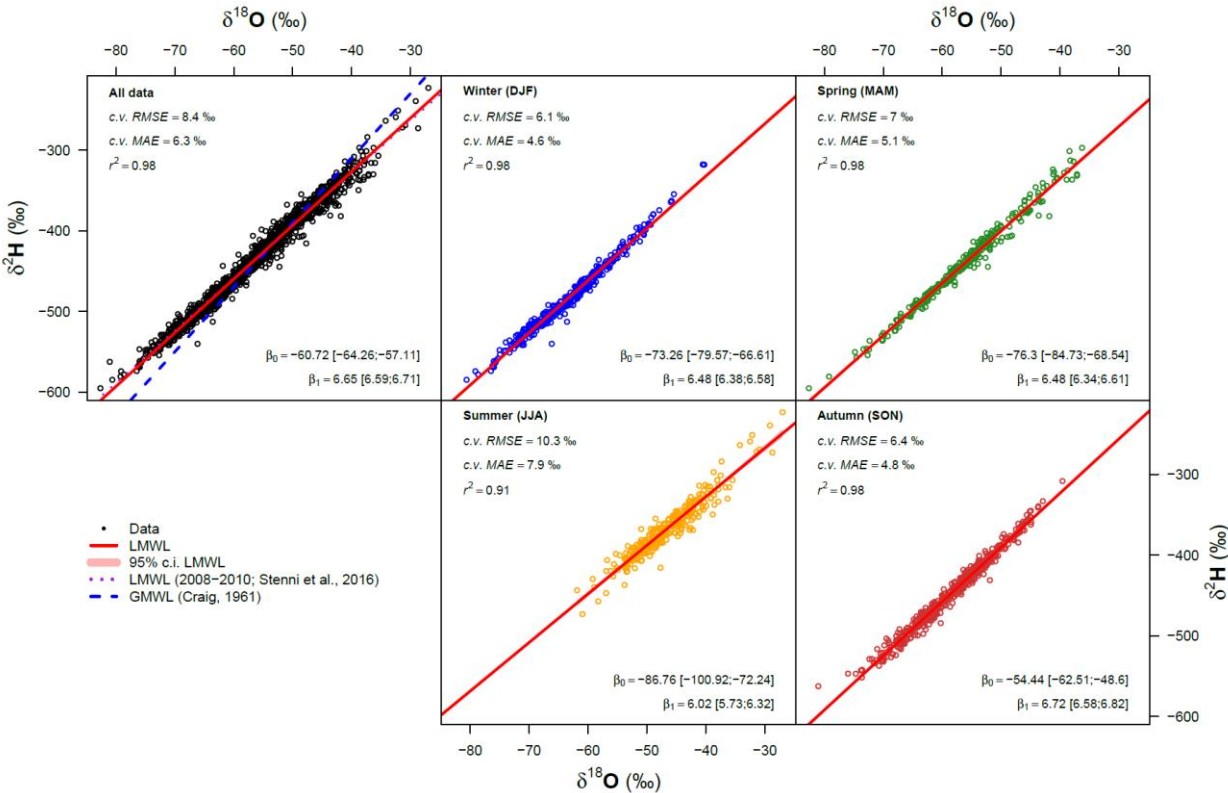

**Figure 3. Local meteoric water lines computed on the isotopic composition of the daily samples for the entire dataset (upper left) and for the single seasons. Regression parameters are also summarized in Table 2. The plot for all data also illustrates LMWLs reported by Stenni et al. (2016) and the global meteoric water line by Craig (1961).**



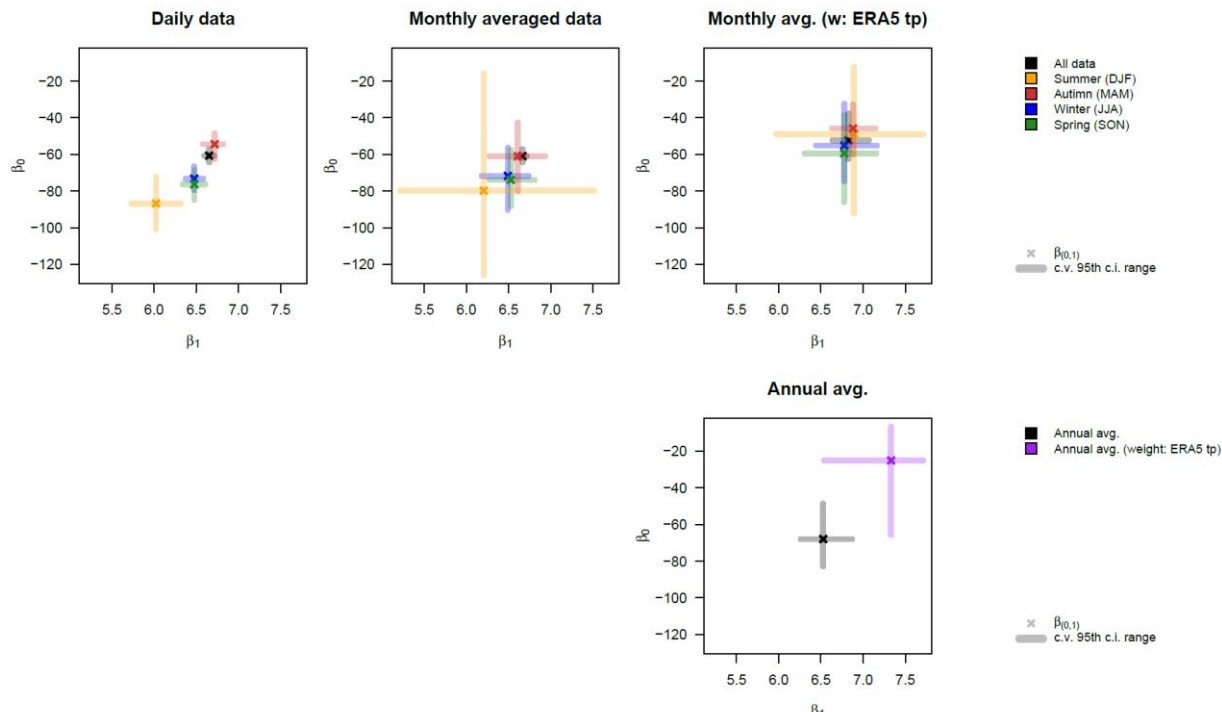

**Figure 4. Intercepts ($\beta_0$) and slopes ($\beta_1$) of all computed LMWLs along with their cross-validated confidence intervals.**





**Table 2. Local meteoric water lines (LMWLs) computed over different time periods.**

| Regr. variables | | Data used | Intercept | | Slope | | $r^2$ | CV RMSE |
|---|---|---|---|---|---|---|---|---|
| DV | IV | | $\beta_0$ (±std. error) | BS 95th CI | $\beta_1$ (±std. error) | BS 95th CI | | ‰ |
| $\delta^2H$ | $\delta^{18}O$ | Daily, all data | -61 (±1) | [-64;-57] | 6.65 (±0.02) | [6.59;6.71] | 0.98 | 8.4 |
| $\delta^2H$ | $\delta^{18}O$ | Daily, W (JJA) | -73 (±3) | [-80;-67] | 6.48 (±0.05) | [6.38;6.58] | 0.98 | 6.1 |
| $\delta^2H$ | $\delta^{18}O$ | Daily, SP (SON) | -76 (±3) | [-85;-69] | 6.48 (±0.05) | [6.34;6.61] | 0.98 | 7 |
| $\delta^2H$ | $\delta^{18}O$ | Daily, SU (DJF) | -87 (±5) | [-101;-72] | 6.02 (±0.11) | [5.73;6.32] | 0.91 | 10.3 |
| $\delta^2H$ | $\delta^{18}O$ | Daily, F (MAM) | -54 (±3) | [-63;-49] | 6.72 (±0.04) | [6.58;6.82] | 0.98 | 6.4 |
| $\delta^2H$ | $\delta^{18}O$ | Monthly, all data | -61 (±4) | [-69;-49] | 6.66 (±0.07) | [6.53;6.87] | 0.99 | 5.9 |
| $\delta^2H$ | $\delta^{18}O$ | Monthly, W (JJA) | -72 (±9) | [-90;-56] | 6.49 (±0.14) | [6.18;6.74] | 0.99 | 3.6 |
| $\delta^2H$ | $\delta^{18}O$ | Monthly, SP (SON) | -74 (±7) | [-88;-58] | 6.53 (±0.12) | [6.28;6.81] | 0.99 | 4.4 |
| $\delta^2H$ | $\delta^{18}O$ | Monthly, SU (DJF) | -80 (±21) | [-126;-16] | 6.21 (±0.44) | [5.21;7.52] | 0.88 | 7.2 |
| $\delta^2H$ | $\delta^{18}O$ | Monthly, F (MAM) | -61 (±9) | [-80;-43] | 6.61 (±0.15) | [6.27;6.93] | 0.99 | 3.4 |
| $\delta^2H_{tp}$ | $\delta^{18}O_{tp}$ | Weighted monthly, all data | -52 (±4) | [-62;-38] | 6.83 (±0.08) | [6.64;7.07] | 0.98 | 6.3 |
| $\delta^2H_{tp}$ | $\delta^{18}O_{tp}$ | Weighted monthly, W (JJA) | -55 (±8) | [-75;-32] | 6.78 (±0.14) | [6.44;7.16] | 0.99 | 3.9 |
| $\delta^2H_{tp}$ | $\delta^{18}O_{tp}$ | Weighted monthly, SP (SON) | -59 (±8) | [-86;-39] | 6.78 (±0.15) | [6.31;7.16] | 0.99 | 5.4 |
| $\delta^2H_{tp}$ | $\delta^{18}O_{tp}$ | Weighted monthly, SU (DJF) | -49 (±18) | [-92;-12] | 6.89 (±0.41) | [5.97;7.71] | 0.91 | 9.3 |
| $\delta^2H_{tp}$ | $\delta^{18}O_{tp}$ | Weighted monthly, F (MAM) | -46 (±7) | [-60;-33] | 6.88 (±0.14) | [6.63;7.15] | 0.99 | 3.1 |
| $\delta^2H$ | $\delta^{18}O$ | Annual | -68 (±9) | [-83;-49] | 6.53 (±0.17) | [6.26;6.87] | 0.99 | 1.5 |
| $\delta^2H_{tp}$ | $\delta^{18}O_{tp}$ | Weighted annual | -25 (±15) | [-66;-7] | 7.33 (±0.29) | [6.54;7.71] | 0.99 | 2.3 |

### 3.4 Deuterium excess

Deuterium excess varied between -47.3 and 85.8‰ (average 15.6‰); the violin plots for the whole period (Figure
S8-9) show quite symmetrical data distributions (median 15.7‰). The seasonal pattern (Figure 2) inversely
followed the air temperature, with higher d-excess is generally recorded during Austral winters (average JJA
22.5‰, average DJF 5.9‰). The average seasonal amplitude of d-excess variations between DJF and JJA was
17‰. The linear relationships between d-excess and $T_{AWS}$ were weaker than for $\delta^{18}O$ and $\delta^2H$ ($r^2$=0.36 to 0.61;
Table 1) with regression slopes of -0.65‰/°C for daily data. This relationship is slightly stronger using monthly
averaged data ($r^2$=0.57) with a slope of -0.65‰/°C and -0.63‰/°C for the monthly averaged and the monthly
weighted averages, respectively. Annually-averaged data show a higher slope (-0.82‰/°C) and an even stronger
relationship ($r^2$=0.61) when using unweighted values.





Deuterium excess exhibits a statistically significant ($p<0.05$) negative linear relationship with $\delta^{18}O$ on daily data ($r^2=0.67$), showing a slope of -1.35 (Figure S20; Table S1), slightly higher than the value of -1.5 reported by
Stenni et al. (2016) for 2008-2010. The linear relationship on daily data is even higher in winter and spring ($r^2=0.75$), but lower in summer ($r^2=0.53$) (Figure S20). Using the monthly unweighted and weighted data, the slopes are respectively -1.34 and -1.18 (Figures S21-22) for the whole dataset. These relationships are summarized in Fig. 5.

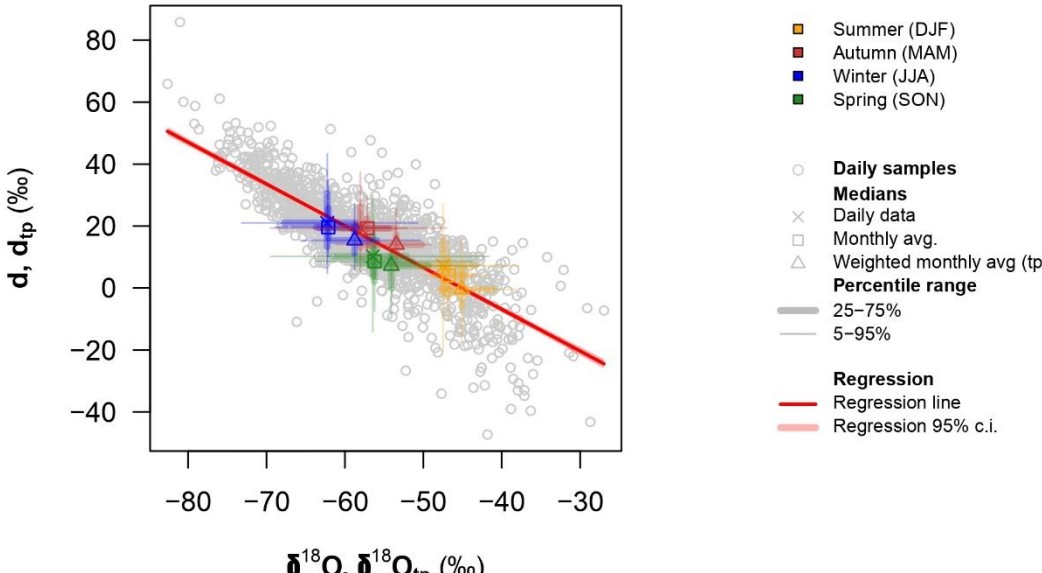


**Figure 5. Relationships between $\delta^{18}O$ and d-excess. The red line represents the linear fit of the data.**

The linear relationship between $\delta^{18}O$ and d-excess computed on the annual-averaged data returns a high coefficient of determination ($R^2=0.91$), however, the one computed on weighted data exhibits a much worse fit
(Figure S23).

An anticorrelation between delta values and d-excess has been already reported in precipitation at Dome C (Stenni et al., 2016) and across continental Antarctica, e.g., at Vostok (Ekaykin et al., 2004) and Dome F (Fujita and Abe, 2006), while it was not observed for coastal areas, e.g., Dronning Maud Land (Schlosser et al., 2008) and Law Dome (Delmotte et al., 2000).

The high d-excess values encountered in winter precipitation and the large seasonal amplitude cannot be only explained with a change in the moisture source regions, but its increase might be related to the very low





condensation temperature and its effect on d-excess, as well as to the decrease in the slope of the MWL at very low temperatures (Touzeau et al., 2016).

The d-excess/$\delta^{18}$O ratios were analyzed to better trace the effects of possible sublimation processes (Figure S24).

Seasonally, the ratios closely follow the pattern of air temperature. While most of the ratio values generally led between 0 and -0.7, positive values are also found during Austral summer and spring. Positive ratios depict negative values of d-excess; extremely positive values, 0.5 up to 1.5, are recorded in the Austral summer.

The d-excess/$\delta^{18}$O relationship was stronger during winter and spring and weaker in summer (Table S1), possibly reflecting the effects of sublimation, due to 24-hour summer solar irradiance, during the permanence of snow on

the benches before sampling. Sublimation effects, acting preferentially during summer, explain the negative values of d-excess mostly found in the summer period (Casado et al., 2021).

### 3.5 Correlations

The daily and monthly-averaged isotope data were analyzed to detect the pairwise correlations with other measured or modeled variables. Since the data distribution of some variables is not normal, the nonparametric

Kendall's rank correlation $\tau$ was computed. Delta values were significantly ($p<0.05$) and moderately ($0.35<\tau<0.6$) to highly ($\tau>0.6$) correlated with air temperature, the temperature of the inversion layer ($T_{INV}$), RH, surface pressure and direct solar radiation, both using daily (Figure S25) and monthly-averaged (Figure S26) data. On the contrary, d-excess was anticorrelated with the temperature of the inversion layer ($T_{INV}$), and RH. Correlations between weather variables reveal positive relationships between air temperature, $T_{INV}$, RH, surface pressure and

direct solar radiation. These relationships were generally observed during the whole 2008-2017 period and singularly during the Austral autumn, winter, and spring, while correlations in summer (DJF) were generally lower. This latter result might depict the effects of the maximum insolation and possible post-depositional processes upon the isotopic composition of precipitation. Under this view, large diurnal cycles in both surface air temperature and humidity in summer may result from either boundary layer dynamics and/or air–snow

sublimation/condensation exchanges (Casado et al., 2016). Generally, SAM was not correlated with any other variable except surface pressure.

No statistically significant ($p<0.05$) long-term linear trends were identified by the Mann-Kendall test for trends during 2008-2017, either using all the monthly-averaged data or analyzing each season separately.



**4. Comparisons with ECHAM5-wiso and ECHAM6-wiso simulations**

The outputs of the ECHAM5-wiso and ECHAM6-wiso model releases are compared with experimental data. In this paper, we mainly focused on ECHAM6-wiso model results while in the SI we also reported the comparison between observations (collected precipitation) and ECHAM5-wiso.

For these comparisons there are some limitations to consider. First, until 2010 samples were collected when at least 5 mL of water equivalent of snow was found on the platform to allow for analysis using the IRMS-

equilibration technique, while smaller samples were collected later due to the availability of the CRDS technique in the laboratory, which requires a smaller amount of water. Thus, the initial experimental data could be representative of higher accumulation events on the bench which could be caused by more intense precipitation events as well as by a significant amount of wind-drifted snow. The samples collected on the platform could also be affected by snow blown either in or out by winds. Finally, although the operators monitored the platform on a

daily basis, post-depositional modification of the isotopic composition of the samples may have happened due to sublimation and/or condensation.

ECHAM5-wiso model simulated 2900 precipitation events in 2008-2017, while ECHAM6-wiso reports 3017, i.e., both models simulated about twice as many precipitation days compared to the experimentally collected samples. Four cases can be observed with respect to the agreement or disagreement between the models and the

experimental observations, as reported in Fig. S27 and S28 (both no precipitation, both precipitation, precipitation only on experimental samples, and precipitation only simulated by models). Figures show that both model versions (ECHAM5-wiso and ECHAM6-wiso) simulated no precipitation when no sample was collected in the platform for about 13% of the days. Conversely, models results imply that precipitation and snow samples were collected during 37% of the period. This result indicates that for half of the period under consideration there was

agreement/disagreement between models and collected snow. In particular, the greatest difference (46% of the period) is found when the models simulate precipitation while no sample was collected at Concordia.

The reason for this difference is not clear. Figures S27 and S28 also report the amount of precipitation estimated by ERA5 in the 4 cases, i.e., when the models and experimental observations agree or not. Precipitation rates from ERA5 are very small for days when both ECHAM-wiso model versions and experimental observations report no

precipitation, while significantly higher values are modeled when both have snow. However, for days when there is a disagreement between models and collected samples, ERA5 predicts less precipitation when only samples have been collected. This could indicate samples affected by snowdrift; however, there are no major differences in wind speed between the 4 groups to support this hypothesis. Evaporation estimated by ERA5 also shows no



strong variation between the 4 cases. Instead, there are significant differences for air temperature, RH, and direct

solar irradiance, all weather variables showing strong seasonality (Figure S3).

Figure 6 shows the time series of the delta values analyzed experimentally and modeled by the ECHAM6-wiso model (the comparison between delta values from ECHAM5-wiso and experimental data is shown in figure SI29). Modeled $\delta^{18}O$ varied between -105 and -25‰ (average -53.8‰) while modeled $\delta^2H$ varied between -938 and -191‰ (average -424‰) in ECHAM6-wiso. Modeled d-excess spans between -97‰ and 86‰ (average 7‰) with

a standard deviation of 10‰, which is 5‰ smaller than the one observed for precipitation. The violin plots in Fig. S8 and S9 show that data distributions were quite wider than the experimental data, especially in the case of ECHAM5-wiso. Overall, the $\delta^{18}O$ and $\delta^2H$ simulated data were less negative than the observations with a larger difference during winter and summer.



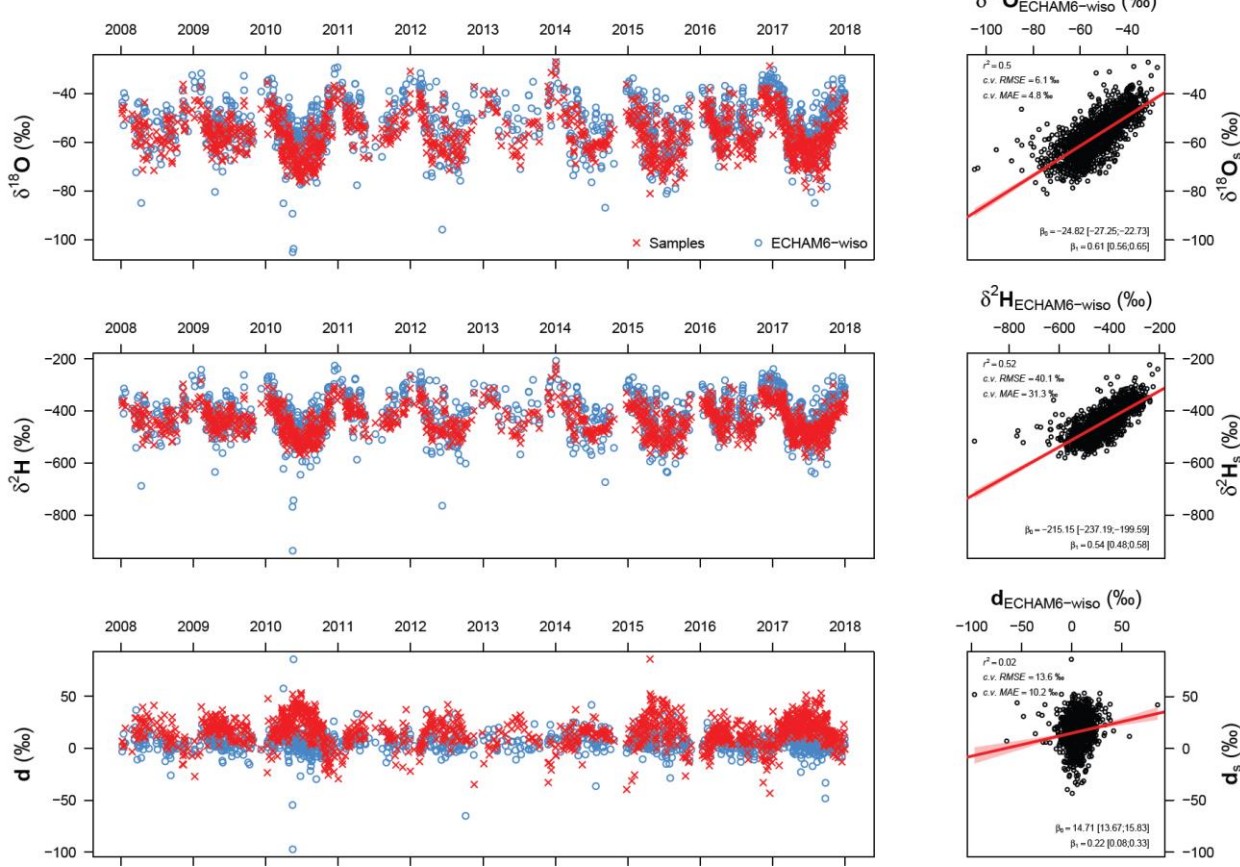

**Figure 6. Time series of the delta values analyzed experimentally (red crosses) and modeled by the ECHAM6-wiso model (blue circles) and their linear regressions.**

The linear relationships between observations and simulated daily data from ECHAM6-wiso are reported in Fig. 6. Moderately good relationships were found for $\delta^{18}O$ ($r^2$=0.5) and $\delta^2H$ ($r^2$=0.52) values, while no relationship was found for d-excess ($r^2$=0.02).

Figure 7 reports the comparison of the seasonal variations of experimentally determined data (observations) and simulated data from ECHAM6-wiso, which confirms the overestimation (less negative values) of the model $\delta^{18}O$ and $\delta^2H$ data compared to observations. Larger overestimations are found in summer for both isotopic ratios, particularly in December and January. For $\delta^{18}O$ alone, a larger overestimation is also found in June (winter). This



overestimation was also present in the simulated $T_{2m}$ compared to $T_{AWS}$, which could in part explain the overestimated isotopic values, although no significant correlation exists between the difference of observed and modeled temperatures vs the observed and modeled isotopic values.

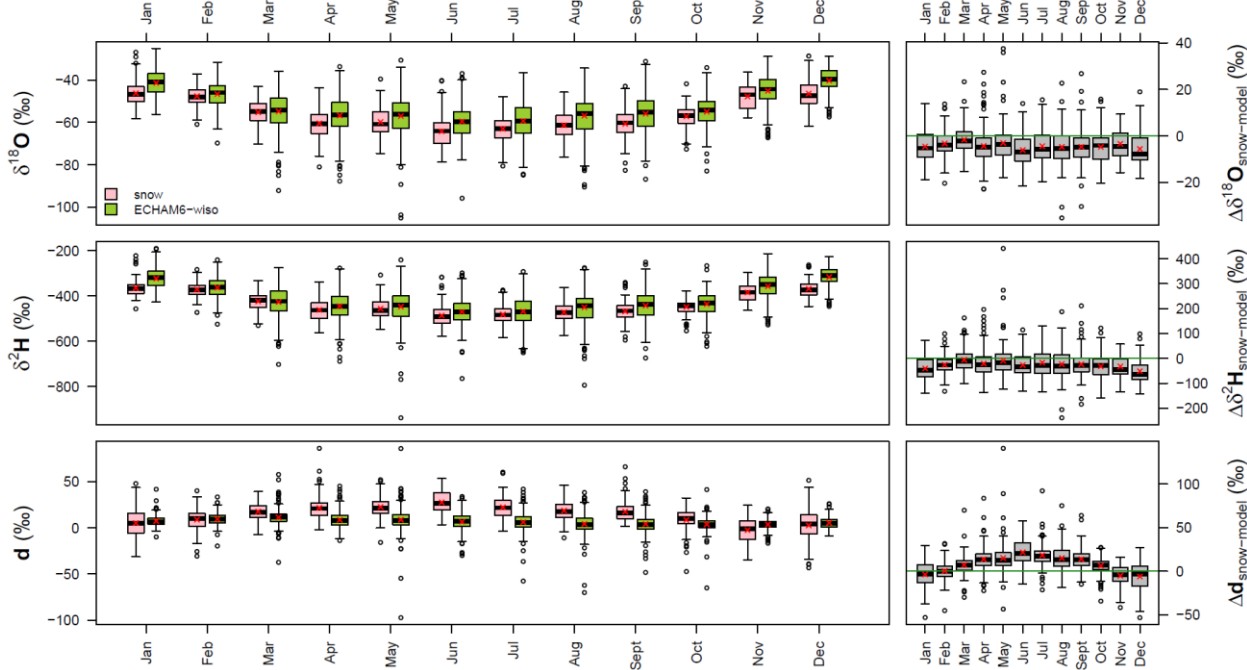

**Figure 7. Seasonal variation of the experimentally measured isotopic data and ECHAM6-wiso modeled data (left) and their differences (right); line = median, box = inter-quartile range, whiskers = ±1.5*inter-quartile range, circles = outliers and extremes; red crosses = arithmetic mean.**





A moderate linear relationship is found between the observed and modeled d-excess vs temperature ($r^2$=0.27), which might indicate the effect of seasonality in model performances simulating kinetic fractionation or non-resolved sub-grid processes.
Indeed, such relationship is more scattered during summer and winter ($r^2$≤0.06) and more robust during spring ($r^2$=0.30).

The local meteoric water lines (LMWLs) computed over different time periods using ECHAM6-wiso outputs are reported in Table S5. The slope ($\beta_1$) obtained from all daily data is 7.66 (±0.02), in good agreement with the one obtained from Masson-Delmotte et al. (2008) for the whole Antarctic surface snow data set (7.75), but slightly higher than the one for the more negative inland values (7.28). On the contrary ECHAM5-wiso simulated lower slope values (Table S3; 6.22±0.03), in better
agreement with observations from this study (6.65 ±0.02).

The simulated d-excess showed the largest discrepancies. Overall, a smaller seasonal amplitude was observed in the d-excess simulated data with the largest discrepancy in winter, which showed lower values than observations; however, the tuning of the supersaturation function could contribute to the d-excess discrepancy.

The relationship between isotopic ratios and temperature simulated by ECHAM6-wiso (Figure S32 and Table S4) exhibited
slightly higher slopes than the observed data. The simple linear relationship of daily values of $\delta^{18}O$ and $\delta^2H$ with $T_{2m}$ was moderate ($r^2$=0.5 and 0.53, respectively). The regression slopes were 0.67‰/°C [cross-validated 95% c.i.: 0.65-0.69‰/°C] and 5.34‰/°C [5.17-5.51‰/°C] for $\delta^{18}O$ and $\delta^2H$, respectively. These positive relationships become stronger on monthly-averaged data ($r^2$=0.88 and 0.9 for $\delta^{18}O$ and $\delta^2H$, respectively) with slopes values of 0.61‰/°C [c.i.: 0.57-0.65‰/°C] and 4.87‰/°C [c.i.: 4.6-5.17‰/°C] for $\delta^{18}O$ and $\delta^2H$, respectively. Finally, regressions computed over annually-averaged unweighted
simulated data exhibited high coefficients of determination but higher slopes (Table S4). At annual basis, regression slopes were 0.87‰/°C [cross-validated 95% c.i.: 0.43-1.13‰/°C] and 7.36‰/°C [4.22-9.36‰/°C] for $\delta^{18}O$ and $\delta^2H$, respectively. Interestingly, if looking at precipitation-weighted annual data these slopes became very low (0.28 and 2.6), but with low coefficients of determinations (0.29 and 0.34), for both $\delta^{18}O$ and $\delta^2H$, respectively.

The results of the comparison between observations and simulated data using the outputs from ECHAM6-wiso are in relatively
good agreement with what previously reported by Goursaud et al. (2018) in their comparison using model outputs from ECHAM5-wiso and field data from the whole Antarctic continent. They also found a warm model bias over central Antarctica as observed in the present study and an overall agreement in the spatial distribution of the isotopic values.

## 5. Conclusions

In this study, we presented a 10-year record of the isotopic composition ($\delta^{18}O$, $\delta^2H$ and d-excess) of daily-collected
precipitation samples at Concordia Station, East Antarctica, from 2008 to 2017; this represents a unique dataset for inland Antarctica.

Despite the difficulties related to collecting samples in such a harsh environment, especially during the Antarctic winter, the daily work of the winter-over personnel of Concordia Station has allowed to build an unprecedented database which will be of extreme importance for interpreting the climate record from oxygen and hydrogen stable isotopes that the Beyond EPICA



1.5 million year ice core will soon provide, as well as other East Antarctic ice core isotopic records. To this end, a comprehensive statistical analysis was performed on our precipitation isotopic data, correlated with the instrumental meteorological records and the outputs from the isotope-enabled ECHAM5-wiso and ECHAM6-wiso model versions.

The bench used to collect precipitation stands one meter from the ground and is shielded by an 8-cm rail on three out of four sides, but wind scouring might still remove part of the accumulation and wind-drifted snow might still contribute to the

deposition collected on the bench, altering the original isotopic composition of precipitation. During summer months, the snow on the bench might also be subjected to sublimation due to the direct solar irradiation for the hours preceding the sampling; this could explain the occurrence of negative d-excess values in this season.

Despite these limitations, the dataset presented in this study is the closest we could get to daily precipitation for a continuous 10-year period at Concordia.

From 2008 to 2017, the Dome C AWS average daily temperature ranged from -82°C to -19°C, with an average value of -53°C. The daily precipitation isotopic data also showed a wide range of variation following a marked seasonality: $\delta^{18}O$ varied between -82.63 and -26.97‰ (average -56.7‰); $\delta^2H$ varied between -595.1 and -223.0‰ (average -438‰); monthly-averaged delta values generally showed the lowest values in June and the highest in January.

A moderately high linear relationship was observed between daily $T_{AWS}$ and $\delta^{18}O$ and $\delta^2H$ ($r^2=0.63$ and 0.64, respectively)

with a slope of 0.52‰/°C for $\delta^{18}O/T_{AWS}$ and 3.52‰/°C for $\delta^2H/T_{AWS}$. The relationships strengthen when using mean monthly averages instead of daily data: $r^2$ increases to 0.82 for both $\delta^{18}O$ and $\delta^2H$, with slopes remaining very similar to daily data: 0.51 and 3.4‰/°C for $\delta^{18}O$ and $\delta^2H$, respectively. Regressions computed over annually-averaged data, despite the elimination of the temperature-isotopes covariance from seasonality, also show similar slopes: 0.59‰/°C and 3.9‰/°C for $\delta^{18}O$ ($r^2=0.75$) and $\delta^2H$ ($r^2=0.76$), respectively.

No statistically significant ($p<0.05$) long-term linear trends were identified during the 2008-2017 period.

The LMWL computed over the entire dataset, as well as those calculated for each season, are characterized by lower slopes compared to the 7.75 value found by Masson-Delmotte et al. (2008) for Antarctic surface snow, and also lower than the 7.28 reported in the same study for $\delta^{18}O$ values below -42.8‰. Even lower slopes are observed in this study when considering only summer (DJF) data, with a slightly worse determination coefficient; this seasonal bias might also be due to the effect of

sublimation.

Deuterium excess displayed an ample interval of variation (from -47.3 to 85.8‰), with a seasonal pattern inversely following air temperature. The negative linear relationship between d-excess and $\delta^{18}O$ is higher in winter and spring, but lower in summer, which could also be attributed to sublimation effects (Casado et al., 2021). The high d-excess values found in winter, as well as its seasonal amplitude, are mostly due to the extremely low condensation temperature rather than to changes in

moisture origin.

The 10-year dataset of the isotopic composition of precipitation presented in this work has allowed us to perform a comparison between the observed and modeled (ECHAM5 and ECHAM6-wiso) $\delta^{18}O$ and $\delta^2H$ of precipitation. ECHAM6-wiso showed, on average, less negative simulated delta values compared to the measured samples, with larger differences during winter,



spring and summer and smaller differences observed in autumn. The linear relationship between observations and simulated

data from ECHAM6-wiso is moderately good for $\delta^{18}$O ($r^2$=0.50) and $\delta^2$H ($r^2$=0.52), while no relationship was found for d-excess ($r^2$=0.02); this is a significant improvement from ECHAM5-wiso for both $\delta^{18}$O and $\delta^2$H, while some issues still seem to affect the d-excess simulation. The different d-excess output from the two model versions could be caused by minor changes in the equation of the supersaturation function, as well as by differences in the modeled influence of the sea ice and the different treatment of wind speed influence on kinetic fractionation during evaporation processes. It is also worth to mention that

ECHAM5-wiso is nudged to ERA-interim while ECHAM6-wiso is nudged to ERA5. ECHAM6-wiso also better captured the amplitude of seasonal variations compared to ECHAM5-wiso. The LMWL computed over ECHAM5-wiso daily data returned a slope of 6.22, while ECHAM6-wiso daily data provided a slope of 7.66, which is similar to the one found by Masson-Delmotte et al. (2009) for Antarctic surface snow (7.75), but higher than the one obtained with measured data (6.65) in this study. However, we have to consider that while ECHAM-wiso models simulate the isotopic

composition of precipitation, the snow we sample on the bench is the result of precipitation combined with possible post-depositional processes such as sublimation and snow blown onto or off the platform by the wind.

To conclude, the dataset presented here represents a unique record, which, together with the meteorological data, will provide a valuable contribution for the comprehension of the mechanisms determining the isotopic composition of precipitation in inland Antarctica and will hopefully benefit the ice core record climatic interpretation in the future. The isotopic composition

of precipitation dataset could also be used as the input of isotope-enabled snowpack models.

The collection of daily precipitation samples at Dome C is still ongoing and the data since 2018 will be presented in future publications, in collaboration with the ITN DEEPICE project. The 2018-2022 dataset will also be compared to surface and sub-surface snow samples at Concordia.

**Data availability**

Isotope data are made available in Zenodo: https://doi.org/10.5281/zenodo.10197160. ERA5 hourly data on single levels from 1940 to present are available through the Copernicus Climate Data Store (CDS) at: https://doi.org/10.24381/cds.adbb2d47 (Hersbach et al., 2023). The Antarctic Meteo-Climatological Observatory AWS Concordia data are available upon request at https://www.climantartide.it/dataaccess/AWS_CONCORDIA/index.php?lang=en (Grigioni et al., 2022). Dome C II AMRC AWS are available at ftp://amrc.ssec.wisc.edu/pub/aws/q1h/. Radiosounding data are available upon request at

https://www.climantartide.it/dataaccess/RDS_CONCORDIA/index.php?lang=it (Grigioni et al., 2022). The Baseline Surface Radiation Network (BSRN) data are available at https://bsrn.awi.de/data/data-retrieval-via-pangaea/. AAO (SAM) index monthly data are available at https://www.cpc.ncep.noaa.gov/products/precip/CWlink/daily_ao_index/aao. The CALVA project and CENECLAM and GLACIOCLIM observatories data are available at https://web.lmd.jussieu.fr/~cgenthon/SiteCALVA/CalvaData.html.




**Author contributions**

GD and BS designed the research. GD, BS and VP performed the lab analysis. MW and AC provided the ECHAM-wiso data. MM performed the data analysis. MM, GD and BS drafted the first version of the paper. DZ, CS, VC, MC, AL, MW, AC, GC and MDG contributed to the writing of the paper. GC provided valuable information on precipitation sampling.

**Competing interests**

The contact author has declared that none of the authors has any competing interests.

**Acknowledgements**

Meteorological dataset and information are achieved by the Italian Antarctic Meteo-Climatological Observatory (IAMCO) https://www.climantartide.it in the framework of the PNRA/IPEV 'Routine Meteorological Observation at Station Concordia' 605 project. We acknowledge using data from: (i) the CALVA project and CENECLAM and GLACIOCLIM observatories; (ii) the Baseline Surface Radiation Network (BSRN, https://bsrn.awi.de/). We also acknowledge Quantarctica and the Norwegian Polar Institute for providing the ETOPO1, IBCSO, and RAMP2 data for basemap usage, the U.S. National Ice Center (USNIC) for providing a shapefile with polygons representing Antarctic ice shelfs and land features, the UK Polar Data Centre, Natural Environment Research Council, UK Research & Innovation for providing the medium resolution vector polygons of the 610 Antarctic coastline. The precipitation measurements at Dome C as well as the isotopic analysis have been conducted in the framework the projects PNRA 2013/AC3.05 (PRE-REC) and PNRA18_00031 (WHETSTONE) of the Italian National Antarctic Research Program "Programma Nazionale di Ricerche in Antartide" (PNRA) funded by MIUR (now MUR). We would also like to thank the logistics staff and winter-over crews at Concordia station for all the investigated periods.

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
