# Peer review of "A decade (2008-2017) of water stable-isotope composition of precipitation at Concordia Station, East Antarctica"

_EGUsphere, 2023_

## Author Comment (AC1)

*General comments*

*This paper reports the last ten years of precipitation isotope observations at Concordia Station, Antarctica. The data set is extremely valuable and should be published. The results are mainly based on correlation analysis and numerous comparisons with reanalysis data and isotopic models.*

**R:** The manuscript was revised taking into consideration your suggestion. We would like to thank the referee #1 for his/her valuable contribution that improved the overall quality of our manuscript. The author's comments are in normal text, the referee's comments are in italic.

*On the other hand, I felt that the scientific novelty is unclear.*

**R:** As also pointed out by referee #2, the dataset presented in this manuscript has several strengths. First, the dataset covers 10 years. This is an unprecedented series of data for daily precipitation in Antarctica and therefore the dataset used here led to more robust results and statistical interpretation. For instance, the inter-annual variability is currently better framed than in the previous paper (Stenni et al., 2016). Second, the current dataset is also more robust for evaluating the isotope-enabled GCMs performance. Under this view, we compared our experimental data with ECHAMs-wiso, but also different models can be evaluated against this dataset. Moreover, our observations are based on precipitation rather than surface snow. Third, the interannual variability is probably better captured in this extended dataset because it is more likely to include atmospheric processes acting on scales larger than 3 years. Fourth, these data can be used as input for isotopic models investigating post-depositional processes of surface snow, firn, and ice core records. Fifth, the basic statistical results, e.g., meteoric water lines, seasonal patterns, weighted values, etc., presented in this manuscript are scaled over different periods, such as daily, monthly, and inter-annual scales. These data may therefore be useful to researchers working on different scientific areas, such as atmosphere, climate and weather.

The aforementioned key points are added to the main text as separate items in order to better elucidate the novelty and added values of this manuscript and dataset (Lines 124-141):

"This dataset represents an unprecedentedly long record of precipitation experimentally measured in East Antarctica with several potential advantages for glaciological and palaeoclimatological studies:
- a better framing of the inter-annual variability of the isotopic composition of precipitation with respect to previous works. Indeed, 10 years of observations more likely include atmospheric processes acting on scales larger than 3 years (Stenni et al., 2016);
- a more robust evaluation of the up-to-date isotope-enabled general circulation models (iGCMs) performances, comparing 10 years of experimental data with outputs from different models. For instance, the data provided in the present study may help to improve cloud parameterization through model-data coupling (e.g., microphysics scheme, ice nucleation rates);
- the experimentally collected precipitation data can be used as input for isotopic models investigating post-depositional processes of surface snow, firn, and ice core records, since the precipitation isotopic composition is the input signal of the atmosphere-snow surface and subsurface systems;
- the basic statistical results, e.g., meteoric water lines, seasonal patterns, weighted values, etc., presented in this study are scaled over different periods, such as daily, monthly, and inter-annual scales. These data may therefore be useful to researchers working on different scientific areas, such as atmosphere, climate and weather. For instance, the data provided in this study may be useful to better constrain the δ-T

thermometer. To this end, the data used in this study are presented as both weighted and unweighted for the precipitation amount.".

*Although the text is clearly written, the results of the correlation analysis are described in detail, and what the results mean is not discussed well. It should be clarified: "What do we learn from this observation?"*

**R:** Part of the answer to this comment is reported in the previous point. We agree that a better discussion of the dataset may be included. This way, we improved the data description and the discussion of the results.

We amended some sections, improving the data discussion: in particular, we included new discussion in Section 3.2 (delta vs. temperature), Lines 344-349, regarding the delta-T relationship, Section 3.3 (LMWL) Line 389 and Section 3.4 (d-excess) Lines 444-456 regarding the relationship between d-excess and delta values.

Lines 344-349: "These regressions parameters show a small variability when separately computed on different years (range 0.32-0.63‰/°C). Thus, the $\delta^{18}$O-temperature slope was almost constant during the 2008-2017, except in 2011 (0.32‰/°C); when excluding 2011, the range was 0.4-0.63‰/°C. This slope range is even smaller than the confidence interval of the interannual slope [0.39;0.83] (Table 1). On the other hand, the slope range variation over 10 years at Concordia station seems to be smaller than the spatial variation (0.6-0.91‰/°C), as reported in Masson-Delmotte (2008).".

Line 389: "This decrease directly impacts the d-excess values (see next section).".

Lines 444-456: "Indeed, as previously reported in Craig (1961) and Uemura et al. (2012), any process which deviates from the average $\delta^2$H-$\delta^{18}$O slope 8 (GMWL) can affect the d-excess parameter. To this end, we calculated the logarithmic version of d-excess to assess whether the observed $\delta^2$H-$\delta^{18}$O of precipitation better fit a curve rather than a straight line (Uemura et al., 2012), as in the canonical definition of d-excess following the GMWL. The logarithmic transformation effectively reduces the sensitivity of the observed d-excess to observed $\delta^{18}$O (slope from -1.35 to -0.58) and almost flattened the sensitivity of the observed d-excess to observed $\delta^2$H (slope from -0.18 to -0.03). Such a smaller sensitivity between δ values and d-excess for the logarithmic transformation highlights first that special attention should be paid when dealing with extremely depleted precipitation since the linear approximation introduced by the GMWL does not hold anymore. This is especially true when attempting to extrapolate any relationship between precipitation d-excess in extremely cold regions and the evaporative conditions of warmer moisture sources. Second, different processes might be involved in the precipitation sample before the collection, such as mixing with wind-drifted snow and sublimation (Ritter et al., 2016), which could translate into a smaller $\delta^{18}$O vs $\delta^2$H slope for precipitation samples. ".

*I think seven figures in the main paper, 34 supplementary figures, and five tables are too much.*

**R:** We strongly believe that the figures and tables reported in the main text are proportional to the information provided and are needed to support the main findings of the research. On the other hand, we

agree that the number of figures and tables in the supplement could be reduced. This way, we have removed 9 figures in the SI, namely former figures S16-S19, S21-S23, S30, S31.

*The manuscript should be revised to reflect the following comments before publication.*

**R:** Please see our point-to-point replies below.

*Major comments*

*(1) L.110-115: I think the issue of the discrepancy between the three years of data (Stenni et al., 2016) and the Antarctic spatial slope is not fully discussed in the main text. The temporal d18O/T slope in this study is smaller than the spatial slope. The cause of this difference and its implications (especially its impact and implications on the recent controversy about temperature reconstruction of the Antarctic ice cores, e.g., Buizert et al., Science, 2021) need to be discussed.*

**R:** We agree with the referee's comments. This comment refers to a long-lasting, controversial, and still unsolved question in paleoclimate reconstructions from Antarctic ice cores. For example, the slopes between the delta values and temperature have been shown to be highly variable considering different time intervals (Casado et al., 2017). The paper by Buizert et al (2021) reconstructed the magnitude of the last glacial maximum cooling using borehole thermometry. A deeper analysis of this issue could be impossible, since we are not considering the post-depositional processes that can act on snow, thus impacting on the isotope-temperature relationship.

We added the reference suggested by the reviewer with a short discussion in the introduction section in order to give a wider view of this issue (Lines 113-119):

"Hence, the slopes between the delta values and temperature have been shown to be highly variable considering different time intervals and locations (Casado et al., 2017). On the contrary, by reconstructing the magnitude of the last glacial maximum cooling using borehole thermometry, Buizert et al. (2021) showed a large variability of the δ-temperature slope considering different ice core locations. Generally, this latter study reported quite higher δ-temperature slopes (range 0.82-1.45 ‰/°C) than studies using water isotope composition. This represents a long-lasting, controversial, and still unsolved question in paleoclimate reconstructions from Antarctic ice cores.".

However, we also need to stress that this manuscript considers precipitation rather than surface snow, firn or ice.

Reference

Casado, M., Orsi, A., and Landais, A.: On the limits of climate reconstruction from water stable isotopes in polar ice cores, Past Glob. Chang. Mag., 25, 146–147, https://doi.org/10.22498/pages.25.3.146, 2017.

*(2) L155-160: Most precipitation isotopic ratios in this study are lower than that of SLAP. In other words, it has been extrapolated. Maybe future studies will be conducted to evaluate the effects of extrapolation, so please describe how many WSs were used (2 or more) and the isotope ratio of each WS.*

**R:** We agree that the isotopic composition of precipitation for some specific events is more depleted than the SLAP composition and hence also more depleted than our most-negative laboratory standard ($\sim$ -424‰ for $\delta^2H$). However, we tested the CRDS analyzer linearity by diluting, with precise weighting, an extremely depleted water ($\delta^2H \sim$ -900‰) with tap water ($\delta^2H \sim$ -56‰). The following figure shows that the instrument response is extremely linear, capturing all the dilution stages with a coefficient of determination that is almost one. Moreover, the true $\delta^2H$ value of the tap water ("known sample") differs only by $\sim$1.4‰ from the calibrated value of tap water using "Calib. STD" line (Calib. STD line was build using standards with $\delta^2H$ ranging from -424‰ to -306‰, shown as red crosses in the plot).

[Figure]

Hence, we are confident that the calibration line for precipitation analysis, as defined by the internal WSs for each analysis run, is valid also outside the span of the WSs and the error on the extrapolated depleted precipitation events is negligible.

Continuing the question related to WSs, we used two standards (one about -300‰ and the other -400‰) for each analysis run to build the calibration line. A third standard (values around -400‰) is used for QC. Note that all the internal standards are calibrated regularly against VSMOW-SLAP.

We edited the text accordingly as follows (Lines 171-175):

"Two working standards were used during each run to build the calibration line and a third working standard was used for quality control. All the working standards are in the range of very negative values as found in Antarctic snow and were regularly calibrated against VSMOW-SLAP. Internal laboratory tests have shown the linearity of the instrumental response outside of the calibration interval."

*(3) Fig. 1: The authors explained that the isotope ratios were measured only when there was enough snowfall to collect data. Specifically, what is the lowest accumulation (mm/day or more)? Based on Fig. 1, the number of isotope data varied considerably from year to year. In particular, 2010 and 2017 are large, and 2015 and 2017 are also somewhat large. Does this mean that the number of days of snowfall events varies significantly from year to year? Is this consistent with the number of days that snow was collected and the amount of snowfall in the reanalysis data?*

**R:** In this dataset, the lowest accumulation values that we observed during collected precipitation (in 2017, which is the only year with these data) ranged between 0.0024 and 0.2126 mm water equivalent (average 0.053 mm water equivalent). However, this value slightly changed from year to year depending on the operator in the field. This is particularly true during the winters when the harshest conditions occur. As observed by the referee, this uncertainty is also shown by the different number of samples collected year by year.

The following figure shows the comparison between the collected precipitation and the ERA5' total precipitation (tp$_{ERA5}$) for 2017. Although the time series for the collected precipitation is only available for one year (2017), there is a good qualitative agreement between the two time series.

[Figure]

*(4) Regarding the discussion on the relationship between d-excess and d18O: Since the slope of D-18O is 6.6 (fig. 3), it is evident by simple mathematics that calculating d-excess with a slope of 8 would be inversely correlated with d18O (Figs. 1 and 5 and others). Relatedly, in Fig. 6, there is a large discrepancy between the model and observed d-excess, but this can also be understood because the slope of the model's D-18O is close to 8 (7.66). However, the authors briefly mention the possible decrease in MWL slope (L. 412-413). I think it is necessary to discuss this point more, as the authors repeatedly showed d-excess vs d18O relationship. Specifically, the logarithmic definition of d-excess (Uemura et al., Climate of the Past, 2012; Markle and Steig, Climate of the Past, 2022) has been proposed to alleviate this problem. At least the impact of the logarithmic definition on these results could be added to the discussion.*

**R:** The logarithmic definition of d-excess effectively reduces the sensitivity of the observed d-excess to observed δ$^{18}$O, yielding a relationship which is like the one predicted by ECHAM6, as shown in the figures below. Similarly to Uemura et al. (2012), the sensitivity of the observed d-excess vs δ$^2$H (slope = -0.18, R$^2$=0.53) is almost flattened after the logarithmic transformation (slope = -0.03, R$^2$=0.05) (data not shown).

[Figure]

However, it is also worth noting that the logarithmic definition of d-excess on ECHAM6 data produces a spurious sensitivity between $\delta^{18}O$ and d-excess (same for $\delta^2H$). Hence, the logarithmic transformation of d-excess better fits the delta for very depleted precipitation but it cannot be used to perform a direct comparison with the model. We argue that some of the reasons for the better fit of the log d-excess to the observations is due to the occurrence of sublimation of precipitation, mixing with wind-drifted snow and/or other post-depositional processes, which are translated into a smaller $\delta^{18}O$ vs $\delta^2H$ slope of precipitation samples.

We believe that we better highlighted this aspect in the revised version of the manuscript (see also previous answer; Section 3,4, Lines 444-456):

"Indeed, as previously reported by Craig (1961) and Uemura et al. (2012), any process which deviates from the average $\delta^2H$-$\delta^{18}O$ slope 8 (GMWL) can affect the d-excess parameter. To this end, we calculated the logarithmic version of d-excess to assess whether the observed $\delta^2H$-$\delta^{18}O$ of precipitation better fit a curve rather than a straight line (Uemura et al., 2012), as in the canonical definition of d-excess following the GMWL. The logarithmic transformation effectively reduces the sensitivity of the observed d-excess to observed $\delta^{18}O$ (slope from -1.35 to -0.58) and almost flattened the sensitivity of the observed d-excess to observed $\delta^2H$ (slope from -0.18 to -0.03). Such a smaller sensitivity between δ values and d-excess for the logarithmic transformation highlights first that special attention should be paid when dealing with extremely depleted precipitation since the linear approximation introduced by the GMWL does not hold anymore. This is especially true when attempting to extrapolate any relationship between precipitation d-excess in extremely cold regions and the evaporative conditions of warmer moisture sources. Second, different processes might be involved in the precipitation sample before the collection, such as mixing with wind-drifted snow and sublimation (Ritter et al., 2016), which could translate into a smaller $\delta^{18}O$ vs $\delta^2H$ slope for precipitation samples."

*(5) If some of the Supplementary figures are to be deleted, the scatterplots of correlations (Fig. S16-23) would be a candidate.*

**R:** Done. See previous points.

*Technical comments*

*L63 "the emprical d-T relathionship valid" -> "…is valid…"*

**R:** Done.

*L75: I think it is better to use some publication (white paper or perspective) instead of a URL as a citation.*

**R:** We add the paper by Parrenin et al., 2017 (The Cryosphere)

*L246 "Figure SI2" -> « Figure S2 »»*

**R:** Done.

L260 "may had led"-> "may have led"

**R:** Done.

L411 "explained with" -> "explained by"

**R:** Done.

L569 "worth to mention" -> "worth mentioning"

**R:** Done.

References:

*Buizert et al., Science, 2021, https://doi.org/10.1126/science.abd2897*

*Markle and Steig, Climate of the Past, 2022, https://doi.org/10.5194/cp-18-1321-2022*

*Uemura et al., Climate of the Past, 2012, https://doi.org/10.5194/cp-8-1109-2012*

**R:** We added two references in the main text (Buizert et al., 2021 and Uemura et al., 2012).

---

## Author Comment (AC2)

*General comments:*

*This manuscript presents a valuable dataset of isotopic compositions of daily precipitation at Concordia station for ten years from 2008 to 2017. The authors did detailed analysis of meteorological conditions, the water isotope data, and model-data comparisons. The dataset is useful to evaluate model performance, investigate climate controls on water isotopes in Antarctic precipitation, and quantify impacts of post-depositional processes on ice core records. However, the structure of the content can be more concise, and the added scientific values of this manuscript, especially compared to Stenni et al. (2016) that presents the first three-year data, are not very clear. Therefore, major revision is suggested before publication.*

**R:** We are grateful to reviewer #2 for his/her comments and for providing input that is useful for improving our manuscript. We believe we addressed the reviewer's comments and highlighted the novelty and the added value of this new study compared to previous published literature. Please see our point-by-point answers for each comment hereafter. The author's comments are in normal text, the referee's comments are in italic.

*Major comments:*

*1, Titles of many subsections are too short to be informative. E.g. "2.2 Analytical ", "3.5 Correlations".*

**R:** We have reformulated some subsection titles as follows:

2.2 Analytical → Water stable isotopes analysis of precipitation samples

2.3 Weather data → 2.3 Weather observations and reanalysis data

2.4 iGCMs → Isotope-enabled general circulation models (ECHAM5- and ECHAM6-wiso)

3.2 Water stable isotope data → Water stable isotope data and its correlation with temperature

3.5 Correlations → Correlations between water stable isotope data and meteorological parameters

*2, The authors stressed in many place that the dataset is 'unprecedented', "unique", or "of extreme importance". However, it is not clear what additional values do this manuscript bring to the research community compared to Stenni et al. (2016). For example, when this dataset is applied to evaluate model performance in Section 4, does it add more confidence in identifying model bias? Does it help to identify a direction to improve the model simulations or for further studies?*

**R:** According to our answer to reviewer #1, the dataset presented in this manuscript has several strengths. Indeed, 10 years of precipitation data led to more robust results and statistical interpretation: e.g. the inter-annual variability of the isotopic signal in precipitation is currently better framed in this study than in the previous paper (Stenni et al. 2016). About the model performances, we describe now why this dataset can be useful, in general, for benchmarking iGCMs (introduction section). More specifically, the comparison between observations and ECHAMx-wiso is possibly biased because of the adopted microphysics

scheme. The data provided in the present study may help to improve cloud parameterization through model-data coupling in d-excess (microphysics scheme, ice nucleation rates…).

The above considerations are now better discussed in the modified "Introduction" section (see also the answer to the referee #1 regarding the scientific novelty).

*3, The authors presented analysis on both weighted and unweighted monthly or annual values. Can the authors elaborate which one is more suitable in which conditions?*

**R:** The weighted and unweighted data and the temporal averaging time strongly depend on the lifetime of the atmospheric processes considered. Generally, weighted data are preferable when they are compared or related to other variables or when comparing different periods. For instance, when comparing intervals in ice core records it becomes clear that each layer archived in the ice is representative of the amount of snow accumulated over a period. This is also true when considering that the precipitation can be distributed unevenly over the year. This impacts the delta-T relationship as well. For example, if all the precipitation in a year occurs in summer, the resulting delta values will be biased towards summer temperatures and will be different from those measured in another year when the precipitation is concentrated in wintertime, although the annual mean temperature could be identical.

Since the dataset presented here may be useful to several scientists for different purposes, we prefer presenting all possible combinations of results to extend as much as possible the usefulness of our data.

We added a sentence in the new paragraph of the "Introduction" section (Lines 137-141) that is already modified for answering to referee #1: "To this end, the data used in this study are presented as both weighted and unweighted for the precipitation amount."

We also added a short sentence in Section 3.2, when we presented some weighted delta values for the first time (Line 320-325):

"The weighted and unweighted data and the temporal averaging time strongly depend on the lifetime of the atmospheric processes considered, a fact particularly important when dealing with precipitation in continental Antarctica, which is unevenly distributed throughout the year (Fujita and Abe, 2006; Turner et al., 2019). Indeed, the weighted $\delta^{18}O$ and $\delta^2H$ values are thought to be better correlated with snowfall temperature (Masson-Delmotte et al., 2008; Servettaz et al., 2023)."

*And why do the authors weight the variables using total precipitation from ERA5 (Line 296), rather than observed precipitation amount?*

**R:** Unfortunately, we do not have those data for the whole period 2008-2017. The quantification of the precipitation amount collected on the benches was made in 2008-2010 and then started again in 2017.

Thus, for the purpose of the present study, we chose to use ERA5' "tp" data. As we discuss in the next point, there is quite a good agreement between the observed and modeled precipitation amount over the year 2017.

As reported in section 2.1, sample collection occurred daily. Thus, the observed precipitation amount analyzed in this study is only a qualitative value that might be related to both fresh snowfall as well as wind-drifted snow and possibly also affected by wind erosion. We better elucidated this concept in section 2.3 (Line 211-213):

"It is worth noting that given the qualitative nature of the observed accumulation, the $tp_{ERA5}$ parameter has been used in this study as representative of the precipitation amount of the observed daily snow samples.".

*If there is no corresponding precipitation in ERA5, how do the authors do the weighting?*

As reported in section 2.1, monthly and annual-averaged weather data were computed only over days with available samples. Hence, the observed precipitation-weighted relationship between isotopes and weather data is only available at monthly and annual timescales. As also reported in the answers to reviewer #1, in the figure below we present a comparison between the collected precipitation and the $tp_{ERA5}$ for 2017. Although this time series is only available for one year, it is possible to see a qualitative good agreement between the two datasets.

[Figure]

Since there is a general good agreement between ERA5 and experimentally-collected amount of precipitation, we are confident that the use of ERA5 data is suitable for the aims of this study.

*4, The authors did extensive correlation analysis between different variables. However, different variables might be correlated because of their common correlation to another variable. For example, the correlation between deuterium excess and temperature discussed from line 388 might be related to their correlation with d18O. It a partial correlation analysis can confirm this point, what is the point of regression analysis between deuterium excess and temperature?*

**R:** We understand this point and agree with the referee. However, since the goal of this manuscript is to provide as much information as possible for other studies, we chose to also add the correlation between d-excess and temperature. In addition, even though a fraction of the correlation between d-excess and

temperature can be explained by the correlation between $\delta^{18}O$ (and $\delta^2H$) it is also important to highlight that d-excess is more sensible to other variables.

*5, The structure and content of the conclusion section can be more concise. For example, the sentences starting from line 527 and line 533 can be shortened, and the sentence starting from line 538 can be removed.*

**R:** Done. The sentences were shortened and removed, respectively.

*Minor comments:*

*Line 22: "AWS", introduce the full form.*

**R:** Done.

*Line 24: 3.5 ‰/°C for δD/TAWS.*

**R:** Done.

*Line 46: "although occurring progressively over successive condensation events between the initial evaporation and the final deposition areas. " This formulation is very strange. What is occurring?*

**R:** Done. We deleted part of the sentence.

*Line 47: "Consequently, the different sensitivity of the empirical δ-T relationship in East Antarctic ice is generally poorly constrained with respect to other regions". It is not clear what this sentence wants to express.*

**R:** Sentence modified. We deleted "with respect to other regions", which was not clear.

*Line 143: "samples in the sealed bags"?*

**R:** Done. Sentence modified.

*Line 268: "daily pattern" not "diel pattern"*

**R:** Done.

*Line 284: "during the days with collected samples" and "during the sampling days" are duplicated.*

**R:** Done.

*Line 485: Are the scatter plots in Fig.6 based on daily values? If the precipitation occurs on different days in simulations and observations, how are they matched together?*

**R:** The scatterplots report only the days with both experimental and modeled precipitation data. This is now also reported in the figure caption.

*Line 537: "this could explain the occurrence of negative d-excess values in this season." Do you have any supporting evidence for this statement?*

**R:** We have no evidence from our study, but this was already observed by other authors in Antarctica, e.g., Casado et al. (2021) and Ritter et al. (2016). We added these two references supporting this evidence.

*Line 545: "mean monthly averages".*

**R:** Done.

*Line 550: Is this based on annual data? Are ten annual data points enough to evaluate long-term trends?*

**R:** No, the linear trend is calculated over the monthly averaged data, thus over 120 points. This was reported in subsection 2.5 "Data processing".

*Line 554: How can sublimation leads to lower slope in LMWL?*

**R:** Ritter et al. (2016) have shown that sublimation processes can cause fractionation in the presence of very porous snow (as in this study, we are not dealing with ice) at the snow–air interface. Moreover, this latter study also reports that "it is possible that snow would behave more like a liquid than like a solid in this respect and would fractionate". Moreover, Sokratov and Golubev (2009), as well as Stichler et al. (2001), previously showed that sublimated snow samples lie on a line with a slower slope than the GMWL.

Stichler, W., Schotterer, U., Fröhlich, K., Ginot, P., Kull, C., Gäggeler, H. and Pouyaud, B., 2001. Influence of sublimation on stable isotope records recovered from high-altitude glaciers in the tropical Andes. Journal of Geophysical Research: Atmospheres, 106(D19), 22613-22620.

Sokratov, S.A. and Golubev, V.N., 2009. Snow isotopic content change by sublimation. Journal of Glaciology, 55(193), 823-828.

*Line 559: "The high d-excess values found in winter, as well as its seasonal amplitude, are mostly due to the extremely low condensation temperature rather than to changes in moisture origin." Do you have supporting evidence for this conclusion statement?*

**R:** This was already shown by Touzeau et al. (2016) and other papers (Craig, 2961, Uemura et al., 2012, etc.), discussing the effects of the decrease of the slope of the meteoric water line at very low condensation temperature. We improved the discussion in the main text, also following the comments of referee #1 (see section 3.4, Lines 444-456):

"Indeed, as previously reported by Craig (1961) and Uemura et al. (2012), any process which deviates from the average $\delta^2H$-$\delta^{18}O$ slope 8 (GMWL) can affect the d-excess parameter. To this end, we calculated the logarithmic version of d-excess to assess whether the observed $\delta^2H$-$\delta^{18}O$ of precipitation better fit a curve rather than a straight line (Uemura et al., 2012), as in the canonical definition of d-excess following the GMWL. The logarithmic transformation effectively reduces the sensitivity of the observed d-excess to observed $\delta^{18}O$ (slope from -1.35 to -0.58) and almost flattened the sensitivity of the observed d-excess to observed $\delta^2H$ (slope from -0.18 to -0.03). Such a smaller sensitivity between $\delta$ values and d-excess for the logarithmic transformation highlights first that special attention should be paid when dealing with extremely depleted precipitation since the linear approximation introduced by the GMWL does not hold anymore. This is especially true when attempting to extrapolate any relationship between precipitation d-excess in extremely cold regions and the evaporative conditions of warmer moisture sources. Second, different processes might be involved in the precipitation sample before the collection, such as mixing with wind-drifted snow and sublimation (Ritter et al., 2016), which could translate into a smaller $\delta^{18}O$ vs $\delta^2H$ slope for precipitation samples.".

---

## Author Response (AR2)

**REVIEWER #1**

*The revised manuscript has improved considerably. Although the focus remains on fundamental statistical analysis of the data set, I think it is well worth publishing.*

*Since the central value of this paper is to provide the precious observational data that will be used in future studies, then providing the data in an easy-to-use format is crucial. However, at the moment, the isotope datasets cited in "Data Availability" are not available for review; reading the Data Availability text alone, it reads as if each data set is uploaded separately. Secondary use would be laborious if this speculation is correct (the authors provide the isotopic data separately). Thus, it is recommended that the data used for the figures in this paper, at least the temperature, the precipitation weighted, and the respective averages used (e.g., daily, monthly, annually), are published in one file in an easily accessible form. It will surely improve the value of this paper.*

**R:** We have prepared an Excel file with all the required data. We have uploaded this file as a new dataset in Zenodo. The dataset includes 3 main datasheets:

1. DAILY data with date, daily dataset for isotopes ($\delta^{18}$O, $\delta^2$H, d-excess), starting and ending date and hour, temperature and RH from the AWS (aws.temp and aws.RH, respectively) and the ERA5 temperature and total precipitation (era5.t2m and era5.tp, respectively);
2. MONTHLY data with months, monthly averages for for isotopes ($\delta^{18}$O, $\delta^2$H, d-excess), monthly averages for isotopes weighted for AWS temperature ($\delta^{18}$O.aws.temp, $\delta^2$H.aws.temp, d-excess.aws.temp), monthly avergages for for isotopes weighted for ERA5 total precipitation ($\delta^{18}$O.era5.tp, $\delta^2$H.era5.tp, d-excess.era5.tp), monthly averages for air temperature and RH from the AWS (aws.temp and aws.RH, respectively) and the ERA5 temperature and total precipitation (era5.t2m and era5.tp, respectively);
3. ANNUAL data with years, annual averages for isotopes ($\delta^{18}$O, $\delta^2$H, d-excess), annual averages for isotopes weighted for AWS temperature ($\delta^{18}$O.aws.temp, $\delta^2$H.aws.temp, d-excess.aws.temp), annual averages for isotopes weighted for ERA5 total precipitation ($\delta^{18}$O.era5.tp, $\delta^2$H.era5.tp, d-excess.era5.tp), annual averages for air temperature and RH from the AWS (aws.temp and aws.RH, respectively) and the ERA5 temperature and total precipitation (era5.t2m and era5.tp, respectively).

The link and DOI of the datasets are provided in the main text (data availability section).

**REVIEWER #3**

*General comments:*

*Thanks for the prompt responses on previous comments. I still have some concerns about the scientific values of this manuscript and the structure.*

*Major comments:*

*Based on the abstract, the main scientific findings are the estimated temporal slope between delta and temperature and the evaluation of ECHAM simulations. However, it is unclear how these findings relate to existing literature and what are the added values. For example, previous studies also reported spatial slopes, is the value derived here similar or different, and what does it mean for ice core interpretation? Previous studies also compared ECHAM5 and 6, what are the added insights here?*

**R:** We thank the referee for the comments. We already replied to his/her concerns in the previous answers regarding the added values of our dataset. The spatial slope is not discussed here but we only pointed out the difference of our temporal slopes from the previous paper (Stenni et al., 2016), which just accounted for 3 years. Moreover, we also discussed the difference between our data and the spatial slope reported by Masson-Delmotte et al. (2008), see Lines 112-115. We also deeply discussed this question further in the introduction.

Previous studies did not compare ECHAM5-wiso or ECHAM6-wiso with precipitation data collected at Concordia station during a 10 year period, which are provided here. Goursaud et al. (2018 CP, https://doi.org/10.5194/cp-14-923-2018) used ECHAM5-wiso data only on 3 years precipitation data in Concordia. Thus, the comparison provided in the present paper is more robust because spanning over a longer period (10 years).

*The introduction may not be informative and concise enough for a non-expert in water isotopes.*

**R:** We modified the introduction following the suggestions of the previous referees, lengthening it for explaining the scientific novelty.

*It is suggested to omit not very relevant stuff to highlight the importance of this study. For example, Line 70: Why is it necessary to list the age of ice cores?*

**R:** We added this information for readers that might be non-expert in ice core science and to highlight the importance of water isotope records obtained so far in East Antarctica.

*Line 78: It is true that low accumulation/low temporal resolution and blowing-winds might undermine water isotope interpretation, but why is it important to this study? Line 82: It is true that post-depositional processes affect ice core records. But since this manuscript does not investigate post-depositional processes, is it necessary to list all the processes?*

**R:** The isotopic composition of precipitation is indeed the input signal for postdepositional processes that need to be taken into account for paleoclimatological studies. Furthermore, the precipitation collected in the present study may also be subject to secondary processes and as such these potential processes must be highlighted. We are confident that this is clearly explained in the main text.

*Line 96, it is suggested to add a topic sentence at the beginning to inform the readers what to expect in this paragraph.*

**R:** Done. We have edited the sentence to make it clearer.

*The conclusion seems to be longer than necessary. For instance, Line 582: Is it necessary to introduce wind-drifting in the conclusion?*

**R:** The drifing snow represents one of the limitations of the precipitation sampling and in our opinion is important to highlight it in the conclusions.

*Line 587: This sentence may be more suitable for Data and methods section rather than Conclusion.*

**R:** The same as before.

*Minor comments:*

*Line 46: Does local temperature drive precipitation isotopes? Or is it just an empirical relationship resulting from their common correlations with condensation temperature?*

**R:** We agree with the referee, but the condensation temperature is not easy to estimate. In addition, the site temperature is usually derived for paleoclimatological reconstructions, also considering that a relationship between condensation and surface temperature is observed in Antarctica.

*Line 589-599: These are simply summaries of main statistics. No implications are discussed for ice core community.*

**R:** We agree that the Conclusion section is long and that some data provided here were already presented in the main text. However, we prefer to include some main results in the Conclusion section. Therefore, the paragraph has been shortened by only keeping key findings.

Regarding the implications for the ice core community, we have already discussed this topic in the Introduction section, also considering the long-lasting and still not resolved question on the sensitivity of isotopes to temperature over different time scales.

*"The precipitation isotopic composition and the surface temperature showed a marked seasonal variation over the investigated period with a moderately high linear relationship at the daily scale. The relationship becomes stronger when using monthly averages. The $\delta^{18}O$ (and $\delta^2H$) to $T_{AWS}$ slope of 0.52‰/°C (and 3.52‰/°C) computed on the daily values slightly increases to 0.59‰/°C (and 3.9‰/°C) when computed over annually averaged data, although no statistically significant (p<0.05) long-term linear trends were identified during the 2008-2017 period."*

*Line 600-618: The texts on LMWL and model evaluation should be evaluated. Implications of the LMWL results should be mentioned.*

**R:** This was already done in the results and discussion and is reinforced here in the conclusions. We are confident that the discussion over this topic does not need improvements.

*Line 608: Why is there a positive bias? Is it common among other GCMs?*

**R:** This positive bias in inland Antarctica was already found in Goursaud et al (2018) for ECHAM5-wiso, as also reported in the manuscript, and is consistent with the warm bias observed for temperatures. We are not evaluating other GCMs in this paper, thus we cannot provide a complete answer to the point raised by the referee. We are already planning to compare our data to other isotope-enabled GCMs in future publications.